# Recurrent Hypernetworks are Surprisingly Strong in Meta-RL

**Jacob Beck**
Department of Computer Science
University of Oxford, United Kingdom
`jacob_beck@alumni.brown.edu`

**Risto Vuorio**
Department of Computer Science
University of Oxford, United Kingdom
`risto.vuorio@keble.ox.ac.uk`

**Zheng Xiong**
Department of Computer Science
University of Oxford, United Kingdom
`zheng.xiong@cs.ox.ac.uk`

**Shimon Whiteson**
Department of Computer Science
University of Oxford, United Kingdom
`shimon.whiteson@cs.ox.ac.uk`

## Abstract

Deep reinforcement learning (RL) is notoriously impractical to deploy due to sample inefficiency. Meta-RL directly addresses this sample inefficiency by learning to perform few-shot learning when a distribution of related tasks is available for meta-training. While many specialized meta-RL methods have been proposed, recent work suggests that end-to-end learning in conjunction with an off-the-shelf sequential model, such as a recurrent network, is a surprisingly strong baseline. However, such claims have been controversial due to limited supporting evidence, particularly in the face of prior work establishing precisely the opposite. In this paper, we conduct an empirical investigation. While we likewise find that a recurrent network can achieve strong performance, we demonstrate that the use of hypernetworks is crucial to maximizing their potential. Surprisingly, when combined with hypernetworks, the recurrent baselines that are far simpler than existing specialized methods actually achieve the strongest performance of all methods evaluated. We provide code at `https://github.com/jacooba/hyper`.

## 1 Introduction

Meta-reinforcement learning [Beck et al., 2023] uses sample-inefficient reinforcement learning (RL) to learn a sample-efficient reinforcement learning algorithm. The sample-efficient algorithm maps the data an agent has gathered so far to a policy based on that experience. To this end, any sequential model such as a recurrent neural network (RNN), can be deployed to learn this mapping end-to-end [Duan et al., 2016, Wang et al., 2016]. Such methods are also called *black-box* [Beck et al., 2023].

Alternatively, much prior work has focused on a category of *task-inference* methods that are specialized for meta-RL. A meta-RL algorithm learns to reinforcement learn over a distribution of MDPs, or *tasks*. By explicitly learning to infer the task, many methods have shown improved performance relative to the recurrent baseline Humplik et al. [2019], Zintgraf et al. [2020], Kamienny et al. [2020], Liu et al. [2021], Beck et al. [2022].

Recent work has shown the simpler recurrent methods to be a competitive baseline relative to task-inference methods [Ni et al., 2022]. However, such claims are contentious, as the supporting experiments compare only to one task-inference method designed for meta-RL, the experiments provide additional compute to the recurrent baseline, and the results still show similar or inferior performance to more complicated methods on the majority of difficult domains. In particular, they consider two toy domains and four challenging domains, with RNNs significantly outperformed on two of the four challenging domains, and superior to the single task-inference baseline on only one.

37th Conference on Neural Information Processing Systems (NeurIPS 2023).

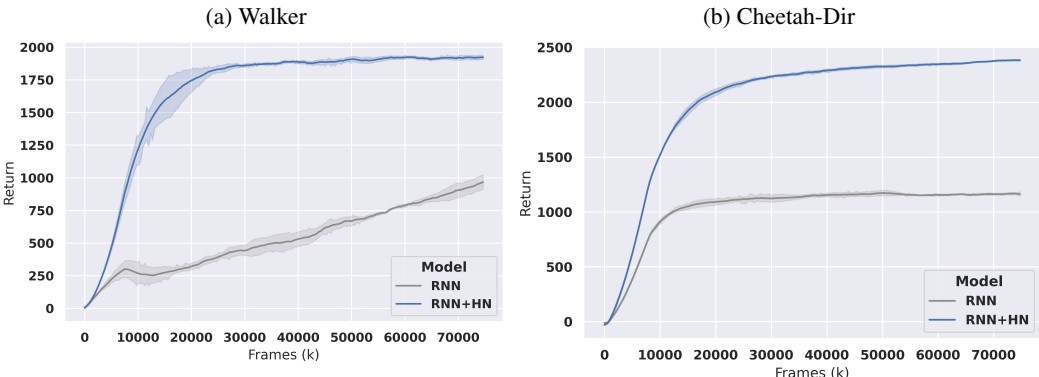

Figure 1: In some environments, recurrent neural networks fail to learn meta-RL tasks, whereas recurrent hypernetworks achieve strong performance.

In this paper, we conduct a far more extensive empirical investigation with stronger and carefully designed baselines in meta-RL specifically. In addition, we afford equal computation in terms of number of samples for hyper-parameter tuning to all existing baselines. We present the key insight that the use of a hypernetwork architecture [Ha et al., 2017] is crucial to maximizing the potential of recurrent networks. For an illustration of the potential magnitude of improvement, see Figure 1. While the use of a hypernetwork with RNNs is not a novel idea, they have never been evaluated in meta-RL beyond a single environment, let alone shown to outperform contemporary task-inference methods [Beck et al., 2022]. We additionally provide preliminary evidence that the robust performance hypernetworks achieve such is in part due to how they condition on the current state and history. Finally, our results establish recurrent hypernetworks as an exceedingly strong method on meta-RL benchmarks that is also far simpler than alternatives, providing significant ease of use for practitioners in meta-RL.

## 2 Related Work

**Recurrent Meta-RL.** Many meta-RL methods structure the learned RL algorithm as a black box using a neural network as a general purpose sequence model [Duan et al., 2016, Wang et al., 2016, Mishra et al., 2018, Fortunato et al., 2019, Ritter et al., 2021, Wang et al., 2021, Ni et al., 2022]. While any sequence model can be used, often the model is structured as an RNN [Duan et al., 2016, Wang et al., 2016, Ni et al., 2022]. Such models [Duan et al., 2016, Wang et al., 2016] are commonly used as simple meta-RL baselines.

One study has shown RNNs to be a competitive baseline in meta-RL [Ni et al., 2022]; however, the scope of the study was broader than meta-RL and the evidence specific to meta-RL is inconclusive. First, the study evaluates only a single specialized meta-RL method [Zintgraf et al., 2020], which was, but is not currently, state-of-the-art [Beck et al., 2022]. Second, the experiments use results or hyperparameters from the original papers, while affording extra computation to tune the RNNs on each benchmark, including dimensions that were not tuned for the other baselines. This computation includes tuning architecture choices, the context length, the RL algorithm used, and the inputs [Ni et al., 2022]. And third, the study does not show particularly strong performance of recurrent methods relative to the chosen specialized baseline. On the MuJoCo domains, the recurrent baseline outperforms the specialized method on only one of these four domains, performs similarly on another, and is significantly outperformed on the other remaining two [Ni et al., 2022]. In contrast, our work compares against four specialized baselines; affords equal computation to all methods, defaulting to hyper-parameters that favor existing task-inference methods for parameters that are not tuned; and still establishes recurrent hypernetworks as the strongest method evaluated.

**Task Inference Meta-RL.** In addition to recurrent meta-RL methods, task-inference methods [Humplik et al., 2019, Zintgraf et al., 2020, Kamienny et al., 2020, Liu et al., 2021, Beck et al., 2022] and policy-gradient methods [Yoon et al., 2018, Finn et al., 2017, Vuorio et al., 2019, Zintgraf et al., 2019] constitute a significant bulk of existing work. We exclude the latter methods from

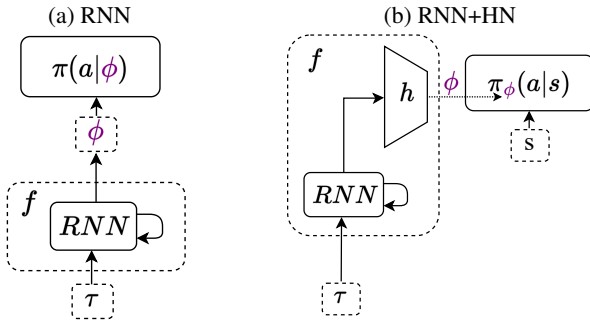

Figure 2: The standard RNN policy (a) and the RNN policy with a hypernetwork (b).

comparison since the estimation of a policy gradient in policy-gradient approaches requires more data than in our benchmarks [Zintgraf et al., 2019, Beck et al., 2023]. Task inference methods are a strong baseline for our benchmark, but are generally more complicated than recurrent meta-RL methods. For example, such methods typically add a task inference objective [Humplik et al., 2019], and may also add a variational inference component [Zintgraf et al., 2021], or pre-training of embeddings with privileged information [Liu et al., 2021]. In this paper, we ablate each of these components to create the strongest task-inference baselines possible. In the end, we find the more complicated task inference methods are still inferior to the recurrent baseline with hypernetworks.

**Hypernetworks.**  A hypernetwork [Ha et al., 2017] is a neural networks that produces the parameters (weights and biases) for another neural network, called the base network. Hypernetworks have been used in supervised learning (SL) [Ha et al., 2017, Chang et al., 2020], Meta-SL [Rusu et al., 2019, Munkhdalai and Yu, 2017, Przewiezlikowski et al., 2022], and meta-RL [Beck et al., 2022, Xian et al., 2021, Peng et al., 2021, Sarafian et al., 2021]. While these networks are complicated, and can fail to work out-of-the-box, simple initialization methods can be sufficient to enable stable learning [Beck et al., 2022, Chang et al., 2020]. In meta-RL, only Beck et al. [2022] have investigated training a hypernetwork end-to-end to arbitrarily modify the weights of a policy. This study suggests that hypernetworks are particularly useful in preventing interference between different tasks and enable greater returns as the the number of parameters increases. However, their study shows a task-inference method to be superior, and the recurrent hypernetwork is evaluated only on a single task with results that are statistically insignificant. Recurrent hypernetworks have never been widely evaluated in meta-RL, let alone shown to outperform contemporary task-inference methods.

## 3   Methods

### 3.1   Problem Setting

A task in RL is formalized as a Markov Decision Processes (MDP), defined as a tuple of $(\mathcal{S}, \mathcal{A}, \mathcal{R}, \mathcal{P}, \gamma)$. At each time-step $t$, the agent inhabits a state, $s_t \in \mathcal{S}$, which it can observe. The agent then performs an action $a_t \in \mathcal{A}$. The MDP subsequently transitions to the next state $s_{t+1} \sim \mathcal{P}(s_{t+1}|s_t, a_t) \colon \mathcal{S} \times \mathcal{A} \times \mathcal{S} \to \mathbb{R}_+$, and the agent receives reward $r_t = \mathcal{R}(s_t, a_t) \colon \mathcal{S} \times \mathcal{A} \to \mathbb{R}$ upon entering $s_{t+1}$. The agent acts to maximize the expected future discounted reward, $R(\tau) = \sum_{r_t \in \tau} \gamma^t r_t$, where $\tau$ denotes the agent's trajectory throughout an episode in the MDP, and $\gamma \in [0, 1)$ is a discount factor. The agent takes actions sampled from a learned policy, $\pi(a|s) : \mathcal{S} \times \mathcal{A} \to \mathbb{R}_+$.

Meta-RL algorithms learn an RL algorithm, $f(\tau)$, that maps from the data, $\tau$, sampled from a single MDP, $\mathcal{M} \sim p(\mathcal{M})$, to policy parameters $\phi$. As in a single RL task, $\tau$ is a sequence up to time-step $t$ forming a trajectory $\tau_t \in (\mathcal{S} \times \mathcal{A} \times \mathbb{R})^t$. Here, however, $\tau$ may span multiple episodes within a single MDP, since multiple episodes of interaction may be necessary to produce a reasonable policy. We use the same symbol, $\tau$, but refer to it as a *meta-episode*. The policy, $\pi_\theta(a|\phi = f_\theta(\tau))$, is parameterized by $\phi$. $f$ is itself parameterized by $\theta$, which are referred to as the *meta-parameters*.

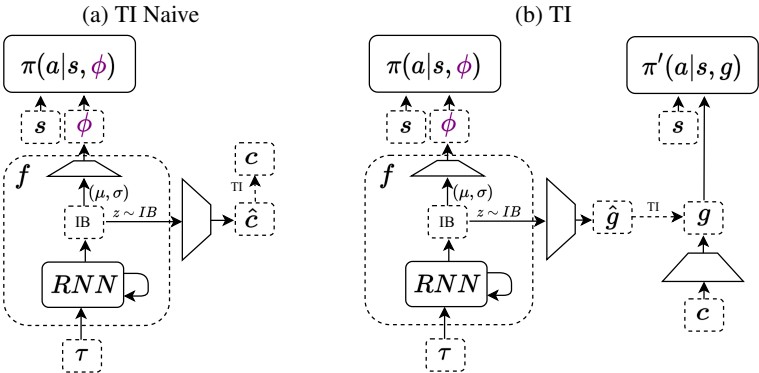

Figure 3: Task-inference baselines. Naive task (a), additional multi-task pre-training (b).

The objective in meta-RL is to find meta-parameters $\theta$ that maximize the sum of the returns in the meta-episode across a distribution of tasks (MDPs):

$$\arg\max_{\theta} \mathbb{E}_{\mathcal{M}\sim p(\mathcal{M})} \left[ \mathbb{E}_{\tau} \left[ R(\tau) \middle| \pi_{\theta}(\cdot | f_{\theta}(\tau)), \mathcal{M} \right] \right]. \tag{1}$$

$f_{\theta}$ is referred to as the *inner-loop*, which produces $\phi$, in contrast to the *outer-loop*, which produces $\theta$.

## 3.2 Recurrent Methods

Recurrent methods are perhaps the simplest and most common meta-RL baseline. Recurrent methods use an RNN to encode history and train all meta-parameters end-to-end on Equation 1. These methods are depicted in Figure 2. While neither recurrent networks (RNN below), nor the the combination of a hypernetwork with recurrent networks (RNN+HN below) is a novel, the combination has never been widely evaluated in meta-RL [Beck et al., 2022], but we will show that the combination actually achieves the strongest results.

**RNN.** Our first recurrent baseline is the simplest and is equivalent to RL2 [Duan et al., 2016] and L2RL [Wang et al., 2016]. In this case $\pi_{\theta}(a|\phi = f_{\theta}(\tau))$, where $f$ is a recurrent network, $\pi$ is a feed-forward network, and $f$ and $\pi$ each use distinct subsets of the meta-parameters, $\theta$.

**RNN+HN.** Our second recurrent model is the recurrent hypernet and achieves the strongest results. Here, the recurrent network produces the weights and biases for the policy directly: $\pi_{\phi}(a|s)$. The state must be passed as input again to this policy for the feed-forward policy to condition on an input, and we follow the initialization method for hypernetworks, Bias-HyperInit, suggested by Beck et al. [2022]. In this initialization, the hypernetwork's final linear layer is initialized with a zero weight matrix and a non-zero bias, so that the hypernetwork produces the same base-network parameters for any trajectory at the start of training.

## 3.3 Task-Inference Methods

Task-inference methods [Beck et al., 2023] constitute the main category of meta-RL methods capable of adaptation as quickly as recurrent (i.e., black-box) methods [Humplik et al., 2019, Zintgraf et al., 2020, Kamienny et al., 2020, Liu et al., 2021, Beck et al., 2022]. These methods train the inner-loop not end-to-end but rather to identify the task, within the given task distribution. One perspective on these methods is that they attempt to shift the problem from the more difficult meta-RL setting to the easier multi-task setting by learning to explicitly infer the task [Beck et al., 2023]. Here we define relevant task-inference methods used as baselines (Figures 3 and 4). For additional ablations, summary, and details on the method selection process, see the appendix.

**TI Naive.** The inner-loop of a meta-RL method must take in the trajectory, $\tau$, and produce the policy parameters, $\phi$. In task inference methods, the inner-loop additionally produces an estimate

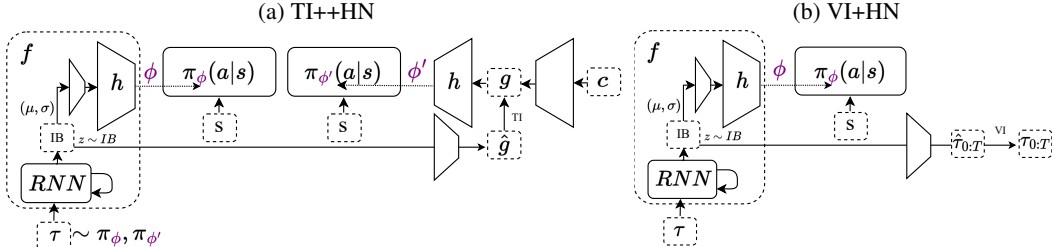

Figure 4: Task-inference baselines. Task inference with additional parameter reuse and hypernetwork (a), an existing contemporary task-inference algorithm [Beck et al., 2022] (b).

of the current task, $\hat{c}_{\mathcal{M}}$, given a known task representation, $c_{\mathcal{M}}$. While it is possible to represent $\phi$ as $\hat{c}_{\mathcal{M}}$ directly, i.e. pass $\hat{c}_{\mathcal{M}}$ to the policy, it is common to compute $\phi$ from an intermediate layer of the network that predicts $\hat{c}_{\mathcal{M}}$, which contains more information about the belief state [Humplik et al., 2019, Zintgraf et al., 2020]. It is also common to use an information bottleneck to remove information not useful for task inference from the trajectory [Humplik et al., 2019, Zintgraf et al., 2020]. Following Zintgraf et al. [2020], we condition $\phi$ on the mean and variance of the bottleneck layer in order to explicitly condition the policy on task uncertainty for more efficient exploration (Figure 3). Putting these together, we can write a task inference method as follows:

$$\mu = P^\mu(RNN(\tau))$$
$$\sigma = P^\sigma(RNN(\tau))$$
$$IB = \mathcal{N}(z; \mu, \sigma)$$
$$\hat{c}_{\mathcal{M}} = P^c(z \sim IB)$$
$$\phi = ReLU(P^\phi \perp (\mu, \sigma))$$
$$J_{infer}(\theta) = \mathbb{E}_{\mathcal{M}}[\mathbb{E}_{\tau|\pi}[-||c_{\mathcal{M}} - \hat{c}_{\mathcal{M}}||_2^2]]$$
$$J_{prior}(\theta) = \mathbb{E}_{\mathcal{M}}[\mathbb{E}_{\tau|\pi}[D(IB||\mathcal{N}(z; \mu = 0, \sigma = I))]],$$

where $P^c$, $P^\phi$, $P_\mu$, and $P_\sigma$ are all linear projections, $\perp$ represents a stop-gradient, $P^\phi \perp (\mu, \sigma)$ is the matrix multiplication of $P^\phi$ with stop-gradient of a concatenation of $(\mu, \sigma)$, and $D$ is the KL-divergence. Here, $J_{infer} + J_{prior}$ constitutes the evidence lower bound from Zintgraf et al. [2020] and is used to train $IB$, whereas all other parameters ($P^\phi$ and $\pi(\cdot|\phi)$) are trained via Equation 1.

**TI.** As presented, the TI Naive baseline may suffer from a known issue where the given task representation contains too little or too much information [Beck et al., 2023]. When too little information is present, the policy may miss information crucial to the task. When too much information is present, the policy may be presented with the difficult problem of separating the useful information from irrelevant task features. Toward this end, it is possible to pre-train the task representation end-to-end using an additional policy [Humplik et al., 2019, Kamienny et al., 2020, Liu et al., 2021]. Our TI baseline (Figure 3) is the same as TI Naive, except that an additional multi-task policy, $\pi'$, is pre-trained to learn a representation of the task, $g_\theta(c_{\mathcal{M}})$. Given a linear projection, $P^g$:

$$\hat{g} = P^g(z \sim IB)$$
$$J_{multi}(\theta) = \mathbb{E}_{\mathcal{M} \sim p(\mathcal{M})}[\mathbb{E}_\tau[R(\tau)|\pi'_\theta(\cdot|g_\theta(c_{\mathcal{M}})), \mathcal{M}]$$
$$J_{infer}(\theta) = \mathbb{E}_{\mathcal{M}}[\mathbb{E}_{\tau|\pi(\cdot|\phi)}[-||g_\theta(c_{\mathcal{M}}) - \hat{g}||_2^2]].$$

For a fair comparison, training of the multi-task policy, $\pi'$, occurs at the expense of training the meta-learned policy $\pi$, with the total number of samples remaining constant. Instead of fully training the multi-task policy, we experiment with different amounts of pre-training in the appendix, finding significant benefits already from less than 5% of total training allocation for the pre-training.

**TI++HN.** The TI++HN baseline is the same as TI, with three additions that we found to strengthen task inference (Figure 4). The first two additions (++) are novel and are 1) initializing the parameters of the meta-policy, $\pi$, to that of the pre-trained multi-task policy, $\pi'$, to encourage transfer and 2)

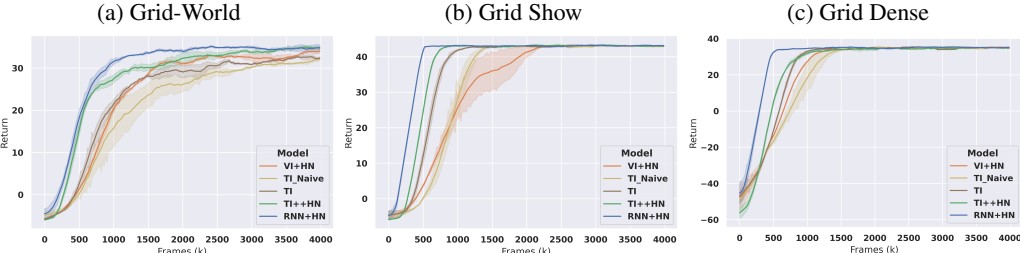

Figure 5: Evaluation on grid-world benchmarks. RNN+HN and TI++HN improve return. RNN+HN achieves the greatest asymptotic return and sample efficiency.

training of the task-inference ($J_{infer}$) over trajectories from the initial multi-task training phase in addition to the meta-learning phase, since the former tend to be more informative and simply provide extra data. The third addition (HN) uses a hypernetwork to condition the policy on $\phi$ [Beck et al., 2022]. We write this as $\pi_\phi(\cdot|s)$ to show that $\phi$ represents the weights and biases of $\pi$, just as in the recurrent baseline. The output of the hypernetwork is $\phi$, and the input to the hypernetwork is a the projection of $\mu$ and $\sigma$, $\phi = h(ReLU(P^\phi\perp(\mu,\sigma)))$. When using a hypernetwork with the first two additions (++), the parameters of the hypernetwork for the meta-learned policy are initialized to the parameters of hypernetwork for the multi-task policy, instead of sharing policy parameters directly.

**VI+HN.** While task-inference methods rely on known task representations, it is also possible to design methods that can infer the MDP more directly. This can be done by inferring transitions and rewards in full trajectories, since the transition function and reward function collectively define the MDP. In particular, such a method, called VariBAD, is proposed by Zintgraf et al. [2020], and extended with the use of hypernetworks by [Beck et al., 2022]. Here, we call this method VI+HN, and it is a contemporary task-inference method (Figure 4). Precisely, this model reconstructs full trajectories including future transitions for meta-episodes, $\tau_{0:T}$, instead of task embeddings, from $\tau$, the current trajectory:

$$J_{infer}(\theta) = \mathbb{E}_{\mathcal{M}}[\mathbb{E}_{\tau|\pi(\cdot|\phi)}[-||\tau_{0:T} - \hat{\tau}_{0:T}||_2^2]].$$

See the appendix for ablations with VI alone.

## 4 Experiments

In this section we compare recurrent hypernetworks (RNN+HN) to task inference baselines. We evaluate over three simple navigation domains [Zintgraf et al., 2020, Humplik et al., 2019, Rakelly et al., 2019], designed to test learning of exploration and memory, in addition to four more difficult tasks using MuJoCo [Todorov et al., 2012], and one task testing long-term memory from visual observations in MineCraft [Beck et al., 2020]. Results show meta-episode return, optimized over five learning rates and averaged over three seeds (four in MineCraft), with a 68% confidence interval using bootstrapping. Additional details and results on Meta-World [Yu et al., 2020] are available in the appendix. Our experiments demonstrate that while our task inference methods are strong baselines, RNN+HN is able to outperform them and achieve the highest returns.

### 4.1 Grid-Worlds

Navigation tasks are a common benchmark in meta-RL [Zintgraf et al., 2020, Humplik et al., 2019, Rakelly et al., 2019]. Here we evaluate on the grid-world variant from Zintgraf et al. [2020], in addition to two of our own variants. The first environment, Grid-World, consists of a five by five grid with a goal location in one of the grid cells. The agent starts in the bottom left corner of the grid, and then must navigate to a goal location, which is held constant throughout the meta-episode. (Details are in the appendix.) This environment is useful for testing how well a meta-learning algorithm learns to efficiently explore in the gridworld as it searches for the goal. Additionally, our Grid-World Show environment was designed to be relatively harder for end-to-end methods, in order to provide a challenge for our proposed method. In this environment, the goal position is visible to the agent

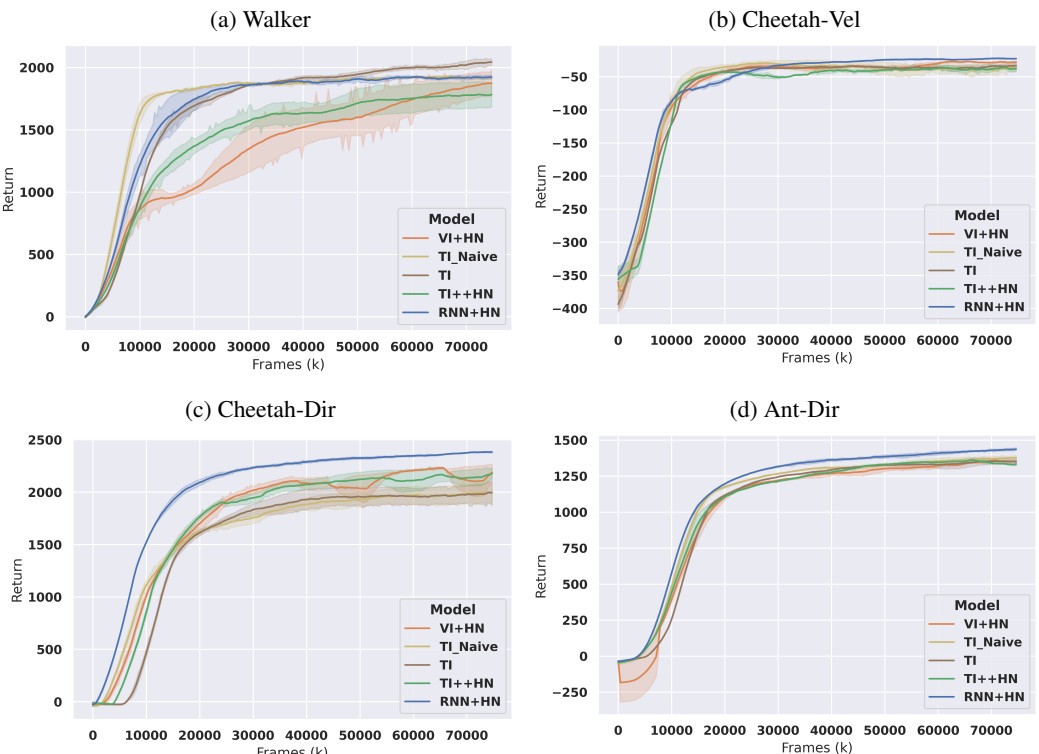

Figure 6: Models evaluated on MuJoCo benchmark. RNN+HN matches returns on Walker and Cheetah-Vel, and exceeds returns on Cheetah-Dir and Ant-Dir.

at the first timestep of each episode. Task inference methods will directly encourage the storage of this information in memory, whereas the end-to-end recurrent methods must learn to store this information through its effect on the policy. In contrast, our Grid-World Dense environment provides dense rewards and may be easier for end-to-end methods. In this environment, the agent receives and observes a reward equal to the Manhattan distance to the goal location. Instead of inferring the task explicitly, the agent can simply move up or to the right until the reward stops increasing.

Surprisingly, on all three grid-worlds, RNN+HN achieves both the greatest asymptotic return and greatest sample efficiency (Figure 5). The recurrent hypernetwork achieves the fastest learning on Gridworld Show, despite the environment being specifically designed to be harder for end-to-end methods. TI++HN dominates all other task-inference baselines on these grid-worlds, suggesting that it is a relatively strong task-inference method. Collectively, these grid-worlds demonstrate that end-to-end learning with hypernetworks can learn to store the task in memory and to explore optimally with this information directly from return, just as well as task-inference methods.

## 4.2 MuJoCo

Here we evaluate baselines on more challenging domains. We evaluate on all four MuJoCo variants proposed by Zintgraf et al. [2020], which is known to be a common and more challenging meta-RL benchmark involving distributions of tasks requiring legged locomotion [Zintgraf et al., 2020, Humplik et al., 2019, Rakelly et al., 2019, Beck et al., 2023]. Ant-Dir and Cheetah-Dir both involve non-parametric task variation, whereas Cheetah-Vel and Walker include parametric variation of the target velocity and physics coefficients respectively. For environment details, see the appendix. We expect Walker in particular to be difficult for end-to-end methods since it has the largest space of tasks with the dynamics defined by 65 different parameters. Assuming the values of these parameters are important for the optimal policy, task inference methods may learn the optimal policy faster.

On Cheetah-Dir and Ant-Dir, RNN+HN achieves greater returns than all other baselines by a wide margin (Figure 6). On Cheetah-Vel, all methods achieve fairly similar results, with RNN+HN

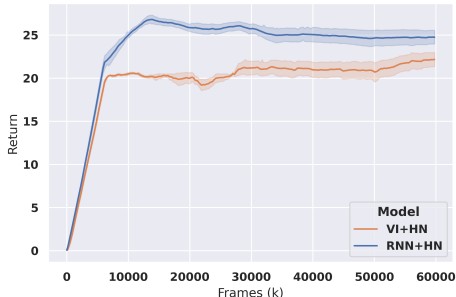

Figure 7: RNN+HN outperforms VI+HN on MC-LS (MineCraft) environment.

still achieving the greatest asymptotic return by a small margin. As expected, RNN+HN does not outperform task inference on Walker. For Walker, only TI outperforms RNN+HN in terms of efficiency, and only TI Naive outperforms RNN+HN in terms of asymptotic return; however, the effect size is small and both TI and TI Naive have among the worst performance on Cheetah-Dir and the grid-worlds. Still, RNN+HN achieves similar performance on Walker, which is notable in a high dimensional task space. And, RNN+HN achieves greater returns overall.

### 4.3 MineCraft

We additionally evaluate on the MC-LS environment from Beck et al. [2020], designed to test long-term memory from visual observations in MineCraft. Here, the agent navigates through a series of 16 rooms. In each room, the agent navigates left or right around a column, depending on whether the column is made of diamond or iron. Discrete actions allow for a finite set of observations. Correct behavior can be computed from the observation and receives a reward of 0.1. At the end, the agent moves right or left depending on a signal (red or green) that defines the task and is shown before the first room. Correct and incorrect behavior receives a reward of 4 and -3, respectively. We allow the agent to adapt over two consecutive episodes, forming a single meta-episode.

On MC-LS we compare RNN+HN to VI+HN alone, given a limited compute budget and since VI+HN is an established contemporary task-inference baseline Beck et al. [2022]. Additionally, we add an extra seed (four in total) and a linear learning rate decay due to high variance in the environment. In Figure 7, we see that RNN+HN significantly outperforms VI+HN. While VI+HN learns to navigate through all rooms, it does not reliably learn the correct long-term memory behavior. In contrast, RNN+HN is able to adapt reliably within two episodes, and one seed even learns to adapt reliably within a single episode. While further work is needed to learn the optimal policy, these experiments demonstrate that RNN+HN outperforms VI+HN, even on more challenging domains.

## 5 Discussion

Here we investigate why the recurrent hypernetworks have such robust performance on meta-RL benchmarks. First, we observe that the state processing differs between the RNN and RNN+HN

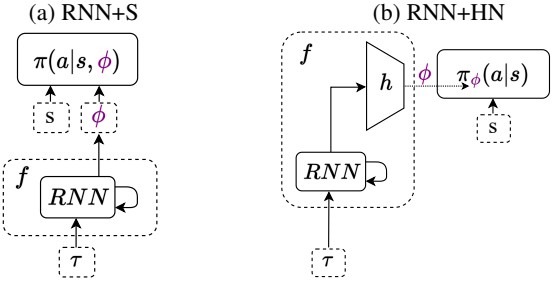

Figure 8: The RNN+S policy (a) and the RNN policy with a hypernetwork (b).

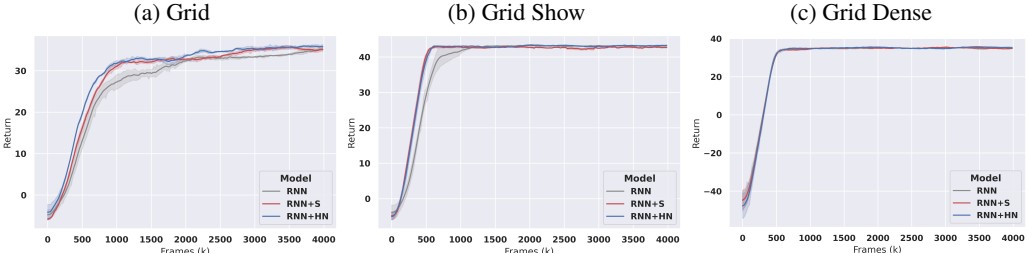

Figure 9: RNN+HN matches or exceeds RNN+S and RNN, but RNN+S is also strong on grid-worlds.

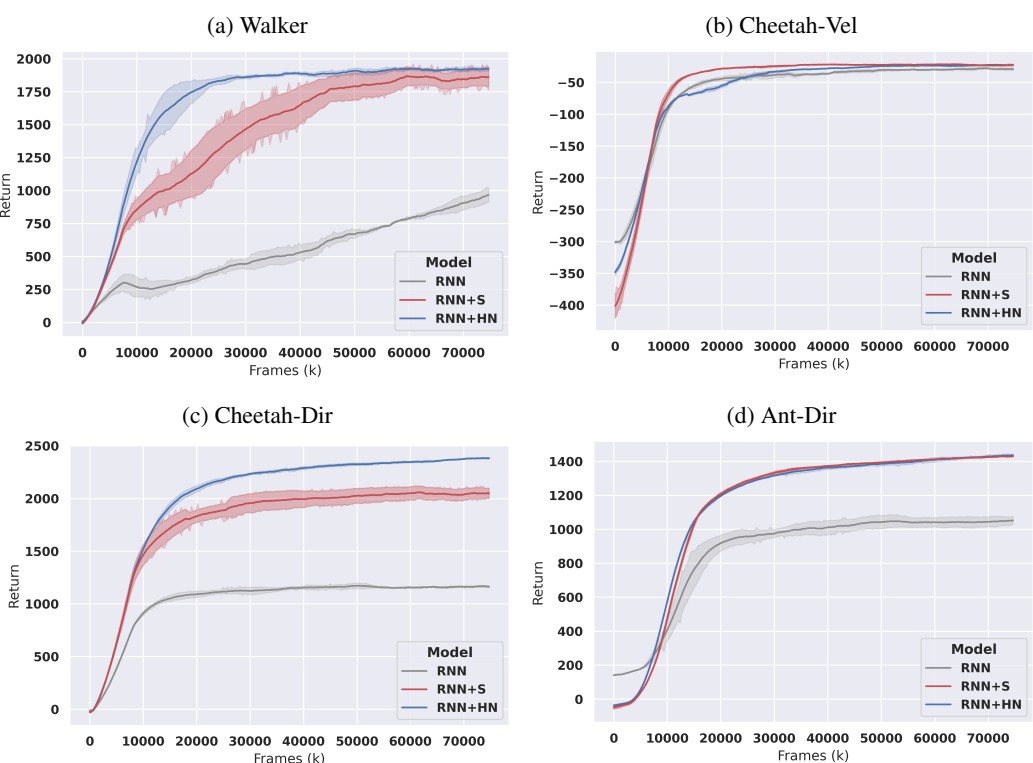

Figure 10: RNN+HN matches or exceeds RNN+S on MuJoCo tasks. RNN alone is a weak baseline.

baselines. In particular, RNN conditions on the current state only through its dependence on $\phi$, whereas hypernetworks pass in the state again to the policy. Thus, we investigate whether the difference in inputs alone could be the cause of the improvement in performance. To this end, we introduce a new ablation to test the effect of just passing in the state again. Details are below. Second, we inspect how sensitivity to latent variables encoding the trajectory affects performance.

**RNN+S.** Hypernetworks condition on the current state both through $\phi$, which contains information about trajectory, including the current state, and by directly conditioning on the state. Since the hypernetwork conditions on state twice, we test to see the effect of conditioning on the state twice without hypernetworks. We call this ablation RNN+S, which we write as $\pi_\theta(a|s, \phi)$ (Figure 8). In an empirical evaluation, we see that while RNN+S does perform favorably relative to RNN alone, RNN+HN still outperforms RNN+S (Figures 9 and 10). In particular, RNN+HN achieves similar returns to RNN+S on Ant-Dir and Cheetah-Vel, and outperforms RNN+S on all other environments, in terms of asymptotic return and sample efficiency. Taken together, we see that the advantage of RNN+HN comes both from the ability to re-condition on state directly and from the hypernetwork architecture. These results confirm that to achieve the strongest performance, re-conditioning on state directly is not sufficient, and that the hypernetwork architecture itself is still critical.

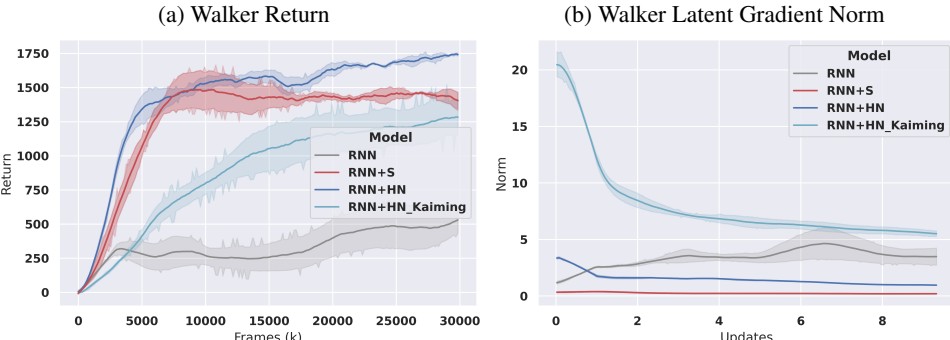

Figure 11: Returns (a) and Gradient norms on Walker (b). The RNN method must increase this norm to condition on state, whereas others do not. Lower gradient norms seem important to performance.

**Latent Gradients.** We also investigate how the hypernetworks condition on the trajectory. In particular, we investigate the sensitivity of the output of the network to an intermediate latent representation of the trajectory. For this purpose, we chose to measure the gradient norm of the first hidden layer of the hypernetwork on the Walker environment. We perform this investigation both with the initialization method designed for hypernetworks that we used throughout our experiments, Bias-HyperInit [Beck et al., 2022], and with Kaiming initialization [He et al., 2015], not designed for hypernetworks. We add Kaiming initialization, since Bias-HyperInit ignores trajectories at the start of training [Beck et al., 2022]. First, we confirm the finding of Beck et al. [2022] that Bias-HyperInit is crucial for performance (Figure 11). Second, we see the two models that performs worst, RNN and RNN+HN Kaiming, also have the greatest norm. Moreover, we find that both RNN+HN and RNN+S start with a low gradient norm and then further decrease this norm throughout training, whereas the RNN model increases this norm. We hypothesize that a low norm, i.e., low sensitivity to the latent variable, is crucial for stable training and that the RNN model increases this norm to remain sensitive to the state, since the state is only encoded in the latent for this model.

# 6   Limitations

As an empirical study of meta-RL, we cannot guarantee that recurrent hypernetworks will improve over every baseline nor on every environment. However, we mitigate this issue by comparing to many baselines and performing many ablations. In particular, we compare to a contemporary task-inference method (VI+HN), design our own baseline which we show to be stronger than others on all grid-worlds (TI++HN), and also include standard methods (TI and TI Naive), in addition to further ablations in the appendix. In as much as an empirical study can, we believe our study demonstrates a significant improvement of the RNN+HN method over existing baselines.

# 7   Conclusion

In this paper, we establish recurrent hypernetworks as a surprisingly strong method in meta-RL. While much effort has gone into designing specialized task-inference methods for meta-RL, we present the surprising result that the simpler recurrent methods can be easily adapted to outperform the task-inference methods. By combining recurrent methods with the hypernetwork architecture, we achieve a new strong baseline in meta-RL that is both robust and easy to implement. In comparison to existing evidence, we provide much stronger empirical results, afford equivalent computation for tuning to all baselines, and establish recurrent hypernetworks as a strong method. We additionally show that passing the state variable to the policy is a crucial component of this method. Finally, we presented gradient analysis suggesting lower latent gradient norms to play an important role in the performance of meta-RL methods. Since the gradient analysis is preliminary and investigates state and latent variables in isolation, future work could investigate the interaction between these variables. Future work could also analyze the interaction between hypernetworks and other sequence models, such as transformers. We hope these insights, along with a simple and robust method, open the way for the broader use of sample-efficient learning in meta-RL and beyond.

## Acknowledgments and Disclosure of Funding

We would like to thank Luisa Zintgraf for her help with the VariBAD code-base along with general advice and discussion. Jacob Beck is supported by the Oxford-Google DeepMind Doctoral Scholarship. Risto Vuorio is supported by EPSRC Doctoral Training Partnership Scholarship, Department of Computer Science Scholarship, and Scatcherd European Scholarship. Zheng Xiong is supported by UK EPSRC CDT in Autonomous Intelligent Machines and Systems (grant number EP/S024050/1) and AWS.

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

# 8 Appendix

## 8.1 Additional Task-Inference Ablations

In order to select task-inference methods for ultimate comparison in the main body, we evaluated those methods, in addition to further task-inference ablations, on the Grid-World and Walker environments, and then selected the best performing methods. From these results, we chose TI++HN (on Grid-World), and TI and TI Naive (on Walker). See Figure 12. We additionally chose to evaluate VI+HN in the main body, as it is a prior strong method Beck et al. [2022]. A table summarizing the components in all methods can be seen in Table 1. The additional ablations are depicted in Figure 13, with details given below:

**VI.** This baseline, called VariBAD, is proposed in Zintgraf et al. [2020] and is the same as VI+HN in the main body, but without the hypernetwork.

**TI++.** This baseline is the same as TI++HN in the main body, but without the hypernetwork. The two additions over TI (denoted "++") are re-use of the multi-task policy parameters as an initialization for the meta-RL policy, and training of task inference over trajectories from pre-training.

**TI+HN.** This baseline is the same as TI in the main body, but with the addition of a hypernetwork.

**BI++HN.** Here, the base-network for the task is inferred (denoted "BI"), instead of the task representation directly. This method is novel and is the same as TI++HN in the main body, but task inference is trained to reconstruct the parameters for the base-network produced by the multi-task hypernet, $\phi'$, rather than the task representation ($c_{\mathcal{M}}$ or $g$):

$$\hat{\phi}' = P^{\phi'}(z \sim IB)$$
$$J_{infer}(\theta) = \mathbb{E}_{\mathcal{M}}[\mathbb{E}_{\tau|\pi}[-||\phi' - \hat{\phi}'||_2^2]].$$

Note that while the performance of BI++HN is similar to that of TI++HN on Grid-World, TI++HN was chosen as a representative baseline since it is more common to existing methods and BI++HN requires significantly more compute.

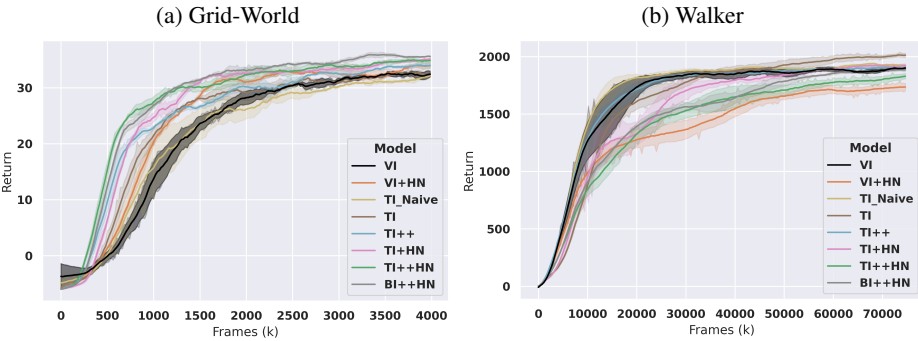

Figure 12: TI++HN was chosen as a baseline given results from Grid-World (a). TI and TI Naive were chosen as baselines given results from Walker (b).

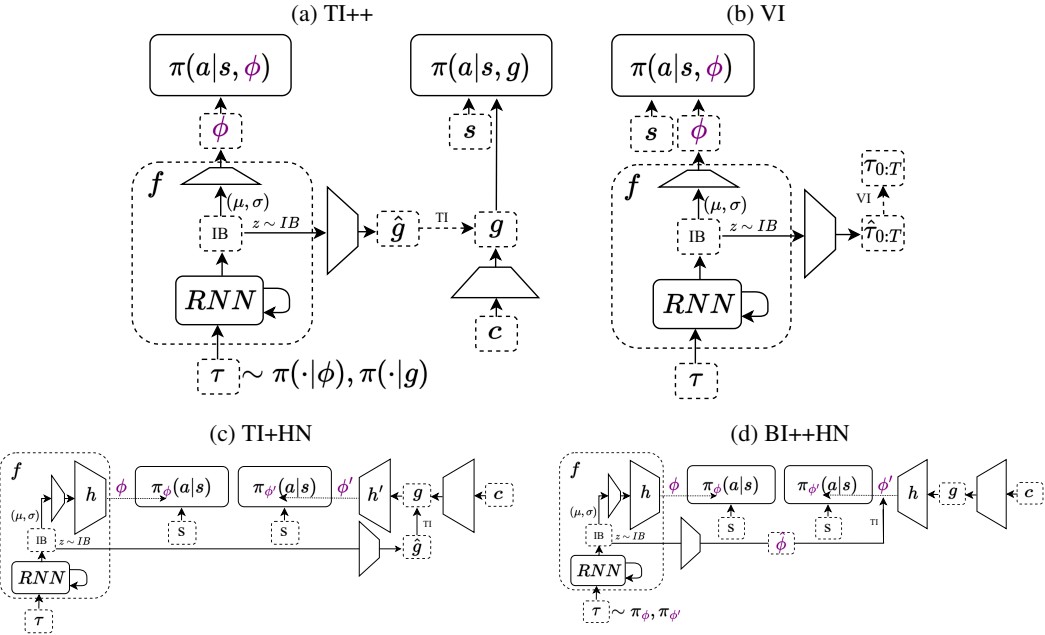

Figure 13: Additional Task-inference baselines.

|  | Inference Target | Policy Conditions on State | Hypernetwork | Inference Training in Multi-Task Phase and Parameter Reuse |
|---|---|---|---|---|
| RNN | None | No | No | N/A |
| RNN+S | None | Yes | No | N/A |
| RNN+HN | None | Yes | Yes | N/A |
| TI Naive | Given | Yes | No | N/A |
| TI | Learned | Yes | No | No |
| TI++ | Learned | Yes | No | Yes |
| TI+HN | Learned | Yes | Yes | No |
| TI++HN | Learned | Yes | Yes | Yes |
| VI | Transitions | Yes | No | N/A |
| VI+HN | Transitions | Yes | Yes | N/A |
| BI++HN | Base Net | Yes | Yes | Yes |

Table 1: Summary of the components in each method

## 8.2 Hyperparameter Tuning

We tune each baseline over five learning rates for the policy, [3e-3, 1e-3, 3e-4, 1e-4, 3e-5], for three seeds each. We use a learning rate for the task inference modules of 0.001, both since varying the learning rate seemed to have little effect (Figure 14) and to provide a fair comparison in terms of computation. Error bars report a 68% confidence interval using bootstrapping.

Before being passed to the hypernetwork or to the policy, depending on the model, the task ($c$ or $g$) are projected down to size 25, as are the concatenation of $(\mu, \sigma)$. This size was chosen to be in the range of the values used by Zintgraf et al. [2020] across environments, but a single value was chosen to be consistent. Additionally, 25 is sufficiently large for a one-hot representation of all discrete task distributions used. However, on Ant-Dir, the projection size is 10. For the state embedding size (passed to the policy) we chose 256 and for the sizes of the MLP policy we chose 256, followed by 128. This corresponds to the "XL" model size in Beck et al. [2022], which was optimal or near optimal where reported. However, for the comparison of RNN, RNN+S, and RNN+HN on Grid-World, these results are reported with a state embedding size for the policy tuned over [5, 25, 125, 625]. For the state embedding size passed to the trajectory encoder, we used 32 for all

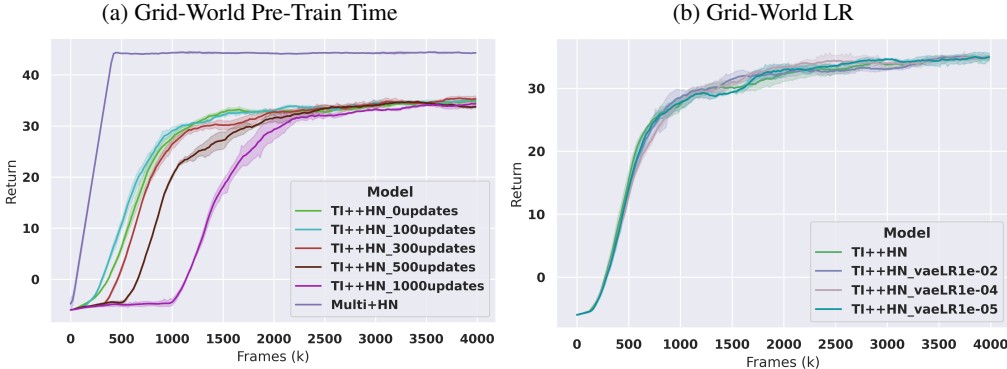

Figure 14: Training for 100 updates yielded optimal performance despite not fully training the multi-task policy, Multi+HN (a). Varying the learning rate for task inference module had little effect (b).

experiments, as the default value in Zintgraf et al. [2020]. And for all RNNs, we use a single GRU layer of size 256, as in Beck et al. [2022].

As discussed in the main body, pre-training of the multi-task policy occurs at the expense of the meta-RL policy training time, in order to ensure the same number of samples for training each method. While selecting the time for pre-training does add a hyperparameter that must be selected, we tune this once on Grid-World and then transfer to all other environments, in order to afford as little additional computation as possible to these baselines. Note that while this cutoff affords some extra computation to task inference methods, the recurrent hypernetwork method (RNN+HN), which we show to be the strongest, does not require any such additional tuning.

On Grid-World, we find 100 PPO updates for training the multi-task policy to be optimal. In this environment, each update consists of 960 frames. Interestingly, while the multi-task policy (Multi+HN) requires approximately 500 updates to fully train, training for 100 yielded optimal performance for TI++HN. (We saw similar results for TI, not reported here, except that the random projection provided by 0 updates already gave similar performance to 100.) Note that 100 updates is equivalent to only 20% of the total frames for training the multi-task policy, and it is only 2.4% of the total frames for training the meta-RL policy. This suggests that very little amount of computation is required to get a good task representation, and delaying the start of meta-training significantly is generally not worthwhile. For MuJoCo experiments in the main body, we use 563 updates, since that is the same percent of total frames (2.4%) for Walker and Ant-Dir, and (4.8% on Cheetah-Vel and Cheetah-Dir), which transferred well on initial experiments.

For all other hyperparameters, we default to those in Beck et al. [2022].

## 8.3 End-to-End and Task Inference Together

While it is possible to combine the end-to-end objective and task inference objective, prior work suggests that interference from competing objectives may be detrimental [Humplik et al., 2019]. Moreover, separating the training objectives can be practically useful, as the combined objectives may require tuning of the relative weighting of the two objectives. Still, we perform investigatory experiments on Grid-World. See Figure 15. We find that adding task-inference objectives to the end-to-end meta-RL objective decreased return compared to the end-to-end objective alone (RNN+HN) for every method tested. The same effect can be seen regardless of the architecture (RNN+HN or RNN+S). Unsurprisingly, the performance of the combined end-to-end and task-inference methods falls in between each type of method used independently. Additionally, tuning the relative relative objective weighting had little effect. We experiment with giving 10% (BI10p, TI10p) and 50% (BI50p) weighting to task inference methods, but still observing the same decrease in return. In all these experiments, we use the RNN architecture from the end-to-end RNN methods, meaning that we did not use an information bottleneck, nor the linear encoding layer. In other words, the output of the RNN passed to the policy, $\phi$, was directly used as $\hat{g}$ or $\hat{c}$ and compared to the respective label $g$ or $c$.

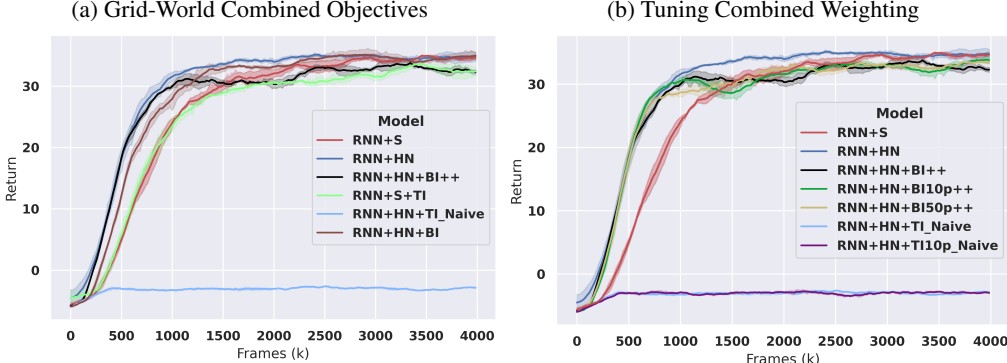

Figure 15: Adding task inference objectives in combination with the end-to-end meta-RL objective decreases performance relative to end-to-end alone, and results in performance between the two types of methods (a). Varying the relative objective weighting had little effect (b).

## 8.4    Meta-World Experiments

We additionally evaluate on Meta-World (ML10) Yu et al. [2020] experiments and show RNN+HN is able to match the performance of the more complicated VI+HN method (Figure 16). Meta-World is a meta-RL benchmark for robotic manipulation. There are 10 non-parametric training tasks and 5 distinct non-parametric test tasks, ranging from pushing a ball to opening a window. Additionally, there is non-parametric variation, e.g. of the goal location, within each task. In this environment, the variance of returns and the computation requirements for training are dramatically greater, so the evidence is not strong is any direction. However, we include the results here since ML10 is a standard benchmark Beck et al. [2023], and since the only experiment ever reported in meta-RL with RNN+HN prior to our paper is on ML10 Beck et al. [2022].

Contrary to our findings, Beck et al. [2022] demonstrated RNN+HN to underperform VI+HN on ML10. However, since that result was not statistically significant, as calculated by Beck et al. [2022], we conduct further experiments on ML10. First, we run more seeds for the VI+HN model in their experiment. (Their experiment is the same as ours, but we use a latent encoding size of 25, instead of 10, for the projection of $(\mu, \sigma)$, to be consistent with the majority of our experiments, and we run for slightly fewer frames due to the intensive computation requirements.) Using different seeds from Beck et al. [2022] resulted in worse performance, indicating that the results truly were insignificant due to variance. Beck et al. [2022] calculate a mean return of 23.9 for VI+HN from the returns: [12.48, 25.69, 33.61]. In contrast, we calculate the final mean return is 18.35 for VI+HN, with is an average over three seeds, with final returns: [32.80, 3.32, 18.93]. Comparing to the returns for RNN+HN from Beck et al. [2022], [11.31, 3.37, 27.77], it is difficult to draw strong conclusion from this experiment.

Additionally, we run our own experiments (with latent size 25) and compare RNN+HN to VI+HN. Contrary to Beck et al. [2022], we find that the final return of RNN+HN is higher than VI+HN. RNN+HN has a final average test success percent of 17.22, compared to 13.87 for VI+HN. However, the variance is still too large to draw firm conclusions. Additionally, we find that the learning curves for RNN+HN and VI+HN overlap consistently throughout training, and the confidence interval for RNN+HN lies almost entirely within that of VI+HN. Note that while tuning hyper-parameters, one of the seeds for the largest learning rate (3e-3) of RNN+HN encountered numerical instability and was ignored. However, 1e-4 was selected as the optimal learning rate, so this is unlikely to affect results, and if it did, would only increase the returns of RNN+HN.

While these results are not sufficient to conclusively show superiority on ML10, we do show superiority on many other environments. Additionally, these results do show RNN+HN to match the performance of VI+HN. Matching existing task-inference baselines on ML10 is significant, since it demonstrates that complicated task-inference methods are not necessary to be competitive.

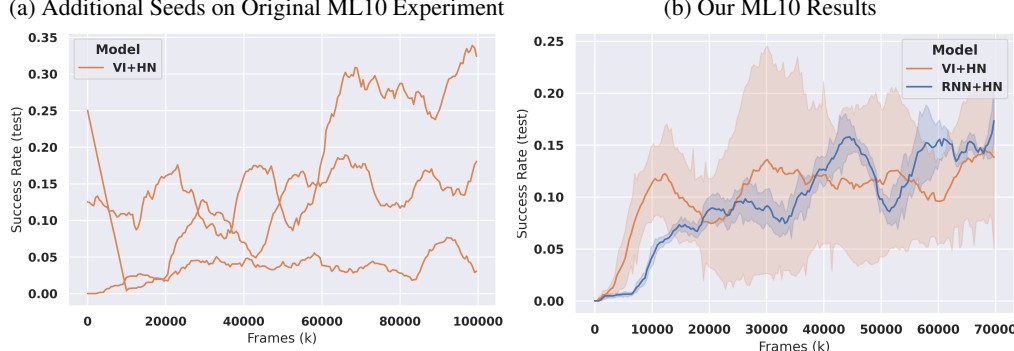

Figure 16: Running three additional seeds of VI+HN on the ML10 experiment from Beck et al. [2022] (with a latent size 10) shows the result is not strong due to high variance. Note that the initial sharp decline in one seed can occur due to smoothing with valid padding on a binary reward (a). While also not a strong result, running our own experiment on ML10 (with a latent size 25) shows that the RNN+HN is able to match the performance of the more complicated VI+HN method (b).

## 8.5 Gridworld Details

Grid-World consists of a five by five grid with a goal location in one of the grid cells. The agent starts in the bottom left corner of the grid, and then has 15 steps in an episode to find the goal and remain there. The agent receives 1 reward for each timestep at the goal and -0.1 reward for all other timesteps. The goal location is held constant for each meta-episode, which consists of 4 regular episodes. In Grid-World Show, the goal position is visible to the agent at the first timestep of each episode. In Grid-World Dense, the agent receives and observes a reward equal to the Manhattan distance to the goal location.

## 8.6 MuJoCo Details

We evaluate on all four MuJoCo Todorov et al. [2012] environments used by Zintgraf et al. [2020]. All environments require legged locomotion. All episodes terminate after 200 timesteps. Meta-episodes in Walker consist of two regular episodes, whereas meta-episodes in the rest consist of one regular episode.

In Cheetah-Dir, the agent controls a robotic cheetah morphology by outputting a control torque for each of six joints on the morphology. The task is to run forward or backward with as large a velocity as possible. The agent observes the position, angle, and velocity of each body part (17 dimensions in total) and is given a positive reward for its velocity in the direction given by the task and a negative reward for control costs (specifically, 5% of the magnitude of the action vector).

In Cheetah-Vel, the agent controls the same robot, but instead of the the positive reward component for running on the selected direction, the agent receives a negative reward component equal to the absolute difference to a target velocity, which is sampled uniformly from [0,3].

In And-Dir, the agent controls a robotic ant morphology by outputting a control torque for each of eight joints on the morphology. The task is to run forward or backward, with as large a velocity as possible. The observations are 27 dimensional in this case, and the agent is given a positive reward for its velocity in the direction given by the task and a negative reward for control costs (specifically, 5% of the magnitude of the action vector). Additionally, the agent is given a negative contact reward (.05% of the magnitude of the external forces on each joint) and a positive survival reward (1 per timestep). Episodes in this environment additionally terminate if the body falls outside of a predefined height.

In Walker, the agent controls a two-legged torso morphology and outputs a control torque for each of six joints on the morphology. The observations are 17 dimensional for this environment. The tasks are defined as uniform samples of 65 different physics coefficients (e.g. body mass and friction) for the simulation. The agent is given a positive reward for the forward velocity, a positive reward of one reward per timestep, and a negative reward for the control costs (specifically, 0.1% of the magnitude

of the action vector). Episodes in this environment additionally terminate if the body falls outside of a predefined height or rotation range.

## 8.7 Compute

Experiments were run on four to eight machines simultaneously each with eight GPUs, ranging from NVIDIA GeForce GTX 1080Ti to NVIDIA RTX A5000. Each model was tuned over five learning rates with three seeds per learning rate. Each experiment took between 4 hours for grid-worlds and 3 days for MuJoCo experiments. In total, there were approximately 360 experiments for MuJoCo and 270 for grid-worlds in the main results. With about 5 experiments on average per GPU, this constitutes roughly 5,400 GPU hours.

