# OpenReview forum: "Recurrent Hypernetworks are Surprisingly Strong in Meta-RL"
_NeurIPS.cc/2023/Conference — NeurIPS 2023 poster_

### Official Review · Reviewer_k7Er · 2023-07-03

**Soundness:** 3 good
**Presentation:** 3 good
**Contribution:** 2 fair
**Rating:** 5
**Confidence:** 3

**Summary:**

The paper investigate the performance of a specific type of architecture, RNNs coupled with hypernetworks, in meta-reinforcement learning. Meta-RL aims to address the sample inefficiency of RL algorithms by learning to perform few-shot learning when given a distribution of related tasks for meta-training.

The authors note that while specialized and complex meta-RL methods have been proposed in the literature, recent work suggests that using an off-the-shelf sequential model, such as an RNN, trained in an end-to-end manner can serve as a strong baseline. However, the supporting evidence for this claim has been limited. The paper presents an extensive empirical investigation to address this gap. While RNNs can achieve strong performance in meta-RL, the study finds that the use of hypernetworks is crucial in maximizing their potential. Interestingly, when combined with hypernetworks, the simpler recurrent baselines outperform existing specialized methods and establish themselves as state-of-the-art (SOTA) on standard meta-RL benchmarks.

**Strengths:**

S1: The paper is extremely clear

S2: The authors make sure to tune each methods, which is something I do not see often enough

S3: Coupling RNNs and hypernetworks is an idea that is novel in the field of meta-RL, and the attempts at understanding why they might be outperforming other baselines are interesting

**Weaknesses:**

W1: While the hyperparameter tuning was fairly done in terms of computation budget, it is to me not clear this strategy is the most relevant. I can imagine that many of the complex methods have many more hyperparameters to tune than the simple methods that you propose, and an excessively small search grid could unfavorably disadvantage the former.

W2: The empirical investigation is, as noted by the authors, quite limited. In that regard, more empirical evidence from different benchmarks would make the paper stronger.

**Questions:**

Q1: Line 128, I could not understand why the predicted context is not directly used to condition the policy. From my understanding, the context should contain enough information to disambiguate the tasks, and the concatenation of the state and context should be Markovian. Could you provide an intuition why intermediate representations are used?

Q2: The RNN-S is an interesting experiment. One additional hypothesis for why RNN+HN might work well is because it allows for some multiplicative interaction between the context and the state. What types of non-linearities were used in the network? Have you used activation units, such as Gated Linear Units (GLU) [1] which allows for multiplicative interactions between the neurons in the RNN-S?

[1]: "Language Modeling with Gated Convolutional Networks", Dauphin 2016

**Limitations:**

The limitations I can think of have been addressed by the authors.

---

> ### Author Rebuttal · Authors · 2023-08-08
>
> Thank you for your feedback. We appreciate you noting the clarity and fair tuning. We address your comments below. If we have addressed these concerns, please do raise our score, and if not, please let us know what remains unclear:
>
> * **1) “While the hyperparameter tuning was fairly done in terms of computation budget, it is to me not clear this strategy is the most relevant. I can imagine that many of the complex methods have many more hyperparameters to tune than the simple methods that you propose, and an excessively small search grid could unfavorably disadvantage the former.”** That is a fair concern, but to mitigate this issue we do use the default parameters from prior work (Zintgraf et al., 2020 and Beck et al., 2022), and we perform additional tuning just for the baselines in the Appendix.
> * **2) “The empirical investigation is, as noted by the authors, quite limited. In that regard, more empirical evidence from different benchmarks would make the paper stronger.”** We do agree that more environments would make our claim stronger and have done so. Please see the global response above.
>
> **Questions:**
>
> * **1) “Line 128, I could not understand why the predicted context is not directly used to condition the policy.… Could you provide an intuition why intermediate representations are used?”** In Humplik et al., 2019 the authors suggest to condition on an earlier representation since the task can be inferred if necessary, but the earlier layer contains more information about the belief state. More specifically, we pass the inferred mean and variance from an information bottleneck layer, as in Zintgraf et al., 2020. This explicit representation of the task uncertainty enables more efficient exploration when the task is not yet known. For example, the uncertainty will inform the agent whether it has to explore more or not. We have now added some clarification of this point into the paper as well.
> * **2) “The RNN-S is an interesting experiment. One additional hypothesis for why RNN+HN might work well is because it allows for some multiplicative interaction between the context and the state. What types of non-linearities were used in the network? Have you used activation units, such as Gated Linear Units (GLU) [1] which allows for multiplicative interactions between the neurons in the RNN-S?”** Beck et al. 2022 did consider FiLM as an alternative and restricted type of multiplicative interactions and found the more general hypernetwork to be more capable. For this reason, we also use the hypernetwork. However, we appreciated the idea and think this could make for good future work!

---

> > ### Author Response · Authors · 2023-08-20
> >
> > Following up on the benchmarks (W2), we now have new experiments that show RNN+HN to match VI+HN on ML10 (in addition to our MineCraft results). Please see the new global comment. As this seemed to be much of the prior concern in this review, if we have addressed this issue, please do consider adjusting the score accordingly.

---

### Official Review · Reviewer_8GEx · 2023-07-04

**Soundness:** 2 fair
**Presentation:** 3 good
**Contribution:** 3 good
**Rating:** 7
**Confidence:** 4

**Summary:**

This paper explores how to approach the problem of meta-reinforcement learning by using hypernetworks. In particular, the authors propose to employ an RL2-like scheme, where instead of producing a vector $\phi$, the recurrent model outputs a set of neural network weights. They dub the method RNN+HN. They introduce a set of task inference-based baselines with and without the usage of hypernetworks. Subsequently, they show that the simple RNN+HN approach outperforms more sophisticated baselines in gridworld environments and in Mujoco. Finally, they perform an ablation study by using RL2 with state-conditioning.

**Strengths:**

Overall, I find the paper interesting and the results quite convincing. However, the authors' claims are very strong -- stating that the proposed method is state-of-the-art in the wide field of meta RL. Although the experiments do show that the method works quite well, I do not think it is enough to support this bold claim. As such, for now, I am going with "borderline reject", but I would be inclined to increase the score if the authors either extended the empirical evaluation or toned down the claims presented in the paper.

Strengths:
- The paper introduces a nice, conceptually simple idea that provides substantial improvements. Parameterizing the policy using a neural network with weights generated by a hypernetwork is an elegant idea.
- The empirical results are in general quite good, the RNN+HN method outperforms most of the baselines in many cases.
- The paper is well written.
- The appendix includes additional ablation studies and more baselines.

**Weaknesses:**

- The critical weakness of the paper is the mismatch between the claims and the empirical evaluation. Namely, the authors state multiple times that their method is state-of-the-art in the field of meta-RL, but the experiments do not support that claim sufficiently:
    - There are not enough external baseline methods. The baselines the authors use are inspired by existing methods but do not correspond exactly to what has been proposed in the literature previously (e.g. TI-naive, TI). Additionally, the authors omit existing established works such as MAML-based algorithms and PEARL. As they say, they "exclude the latter methods since estimation of a policy gradient in policy-gradient approaches requires more data than in our benchmarks". That is, in general, a reasonable assumption, but seems too strict when claiming state-of-the-art.
    - The authors support their claim by saying that their method outperforms previous SOTA [1, 2], but I'm not convinced that these previous works are still SOTA as of now. Some of the recently published works also show very good results [3, 4].
    - Currently, the authors only consider two sets of environments: gridworlds and Mujoco continuous control. I think the empirical evaluation should be extended to include more complex environments such as Meta-World, RLBench, Atari, or DeepMind Alchemy. I think that environments with high-dimensional state spaces (e.g. images) would be interesting to test the limits of the proposed method.
    - There are some environments where RNN+HN falls behind (e.g. Walker, Cheetah-Vel). This is not a grave problem by itself, but it makes the problem of having relatively few benchmarks even more problematic -- how will this method scale up to other environments?
    - Why the RNN baseline only appears in a single environment in Figure 8 and Figure 9? I think it's an important baseline that should be included.

[1] Zintgraf, Luisa, et al. "Varibad: A very good method for bayes-adaptive deep rl via meta-learning." arXiv preprint arXiv:1910.08348 (2019). \
[2] Beck, Jacob, et al. "Hypernetworks in meta-reinforcement learning." Conference on Robot Learning. PMLR, 2023. \
[3] Melo, Luckeciano C. "Transformers are meta-reinforcement learners." international conference on machine learning. PMLR, 2022. \
[4] Chalvidal, Mathieu, Thomas Serre, and Rufin VanRullen. "Meta-Reinforcement Learning with Self-Modifying Networks." Advances in Neural Information Processing Systems 35 (2022): 7838-7851.

**Questions:**

My main suggestion related to the first point in the Weaknesses section above, i.e. extending the empirical evaluation or reducing the state-of-the-art claims.

**Limitations:**

The author discuss limitations sufficiently.

---

> ### Author Rebuttal · Authors · 2023-08-08
>
> Thank you very much for your feedback. We especially appreciate you noting the results are quite convincing and that you are open to increasing the score if we adjust our claims or add additional experimentation. We have now done both, and would very much appreciate a corresponding score increase. Answers to your specific questions are below. Please let us know if anything remains unclear:
>
> * **1) “There are not enough external baseline methods… the authors omit existing established works such as MAML-based algorithms and PEARL”** VariBAD (Zintgraf et al., 2020) already establishes a performance improvement over PEARL and MAML on the same domains we use. For this reason, we compare to VariBAD, a contemporary VariBAD with a hypernetwork (Beck et al., 2022), and direct task-inference methods. It is fair to note that the direct methods, e.g. TI and TI Naive, are not identical to previous methods. However, these were chosen to maintain the strengths of both VariBAD while also using the more direct supervision afforded by the Belief Agent (Humplik et al., 2019). We feel it is actually more fair to do so because the combination uses insights from both papers, and it makes hyperparameter tuning more fair.
> * **2) The authors support their claim by saying that their method outperforms previous SOTA [1, 2], but I'm not convinced that these previous works are still SOTA”** The difficulty in establishing SOTA with hundreds of meta-RL papers (Beck et al., 2023) is well taken. We have now adjusted all of our claims throughout the paper to assert that recurrent hypernetworks are surprisingly strong instead of SOTA. The goal of this paper is to show that simpler recurrent methods (trained end-to-end) can vastly outperform contemporary alternatives (often considered SOTA) in the task-inference design space, when hypernetworks are used. The use of transformers (Melo et al., 2022), as suggested, is an orthogonal consideration which we do agree could further improve the latent variable passed to the hypernetwork. This could make for great future work.
> * **3) “Currently, the authors only consider two sets of environments: gridworlds and Mujoco continuous control… I think that environments with high-dimensional state spaces (e.g. images) would be interesting”** We do agree that more environments (including image observations) would make our claim stronger and have done so. Please see the global response above.
> * **4) “There are some environments where RNN+HN falls behind (e.g. Walker, Cheetah-Vel).…how will this method scale up to other environments?”** Note that on Cheetah-Vel, TNN+HN still has the greatest return at both the start and end of training. As for Walker, the only baselines (TI and TI_Naive) that outperform RNN+HN only do so marginally and are significantly outperformed by RNN+HN on the vast majority of environments. For scaling up, please see the global response above.
> * **5) “Why the RNN baseline only appears in a single environment in Figure 8 and Figure 9? I think it's an important baseline that should be included.”** We have since run these experiments and results confirm that the RNN is in fact always performs the same or worse than RNN+S and RNN+HN, as expected. Please see the global response above.

---

> > ### Comment · Reviewer_8GEx · 2023-08-14
> >
> > Thank you for the thorough response, the modifications introduced to the paper, and the additional experiments. Given that the SOTA claims were removed and most of my other comments were addressed, I decided to increase my score to 5: Borderline Accept. I was thinking about increasing the score further, but I decided against it for now, given the comments from the other reviewers. In particular, I find Reviewer YdBr's argument about the Meta-World results from a previous paper convincing. I will follow the further discussions and update my score accordingly.

---

> > > ### Author Response · Authors · 2023-08-14
> > > **Response to 8GEx**
> > >
> > > Thank you very much for the current and ongoing score consideration. One important point on the Meta-World experiments, for clarification, is below.
> > >
> > >   * The Meta-World experiment in Beck et al., 2022 is not statistically significant, which makes it a bit misleading. The fact that that singular results carries so much weight is one of the reasons that the much stronger countervailing evidence in our paper is so necessary to the conversation.

---

> > > > ### Author Response · Authors · 2023-08-20
> > > >
> > > > Following up on Meta-World, we now have new experiments that show RNN+HN to match VI+HN on ML10 (in addition to our MineCraft results). Please see the new global comment. As this seemed to be much of the prior concern in this review, if we have addressed this issue, please do consider adjusting the score accordingly.

---

> > > > > ### Comment · Reviewer_8GEx · 2023-08-21
> > > > >
> > > > > Thank you for your response and for the additional experiments. It's unfortunate that you cannot show us the plots from the ML10 training, but I do understand this is due to the limitations of the NeurIPS reviewing process and I trust that you describe the results faithfully.
> > > > >
> > > > > I re-read the whole discussion here and I don't find any worrying issues with the authors' submission anymore. The only potential problem in my mind is the amount of changes done throughout the rebuttal process, as there were quite a few. However, I still think the paper has not changed enough to warrant another review process, and as such I decided to increase my score to 7: Accept.
> > > > >
> > > > > At the same time, my decision relies on some promises made by the authors that cannot be verified now (e.g. removing the claims about SOTA from the whole paper). I trust that in the case this paper is accepted, the authors will implement the changes as promised.

---

### Official Review · Reviewer_rCPG · 2023-07-07

**Soundness:** 4 excellent
**Presentation:** 4 excellent
**Contribution:** 3 good
**Rating:** 6
**Confidence:** 3

**Summary:**

This paper investigates the performance of recurrent neural networks in meta-RL. They suggested that a current neural network can achieve strong performance with hypernetwork. They compared this method with numerous baselines on several meta-RL benchmarks and found that the recurrent baselines along with hypernetwork could achieve SOTA performance.

**Strengths:**

This paper introduced a novel assertion that a recurrent method, when combined with a hypernetwork, can achieve SOTA performance in meta-reinforcement learning. The authors substantiated this claim with rigorous empirical experiments demonstrating superior performance. Furthermore, they conducted an analysis elucidating the reasons behind the method's impressive performance. The paper is well-written and clearly structured, which makes it easily comprehensible for the readers.

**Weaknesses:**

1. The authors could enhance the comprehensiveness of the study by testing their algorithm on a wider range of environments.

2. The final analysis segment of the paper would benefit from further development to unequivocally ascertain why the proposed method can achieve state-of-the-art performance.

3. The paper lacks ablation studies for the different settings of hypernetwork component, leaving its individual contribution to the overall results unclear.

4. The format of the reference is wrong

**Questions:**

1. Why in the Cheetah environment, RNN+HN is worse than RNN+S?

2. Please offer more ablation studies of the hypernetwork component.

**Limitations:**

The authors have discussed the limitations of this paper in the last section.

---

> ### Author Rebuttal · Authors · 2023-08-08
>
> Thank you very much for your feedback and for noting the rigorous empirical experiments, elucidating analysis, and clear structure. We address each of your points in turn below. Please do  adjust the score correspondingly, or let us know if anything remains unclear. Thank you!
>
> * **1) “The authors could enhance the comprehensiveness of the study by testing their algorithm on a wider range of environments.”** We do agree that more environments would make our claim stronger and have done so. For details, please see the global response above.
> * **2) “The final analysis segment of the paper would benefit from further development to unequivocally ascertain why the proposed method can achieve state-of-the-art performance.”** In addition to preventing interference between different tasks and scaling trends suggested by Beck et al., 2022, we hypothesize that the low norm afforded by our models, i.e., low  sensitivity to the latent variable, and re-conditioning on state are crucial for stable training. We have added information on existing motivation for the use of hypernetworks in related work and on our analysis in the introduction to make this paper more self-contained.
> * **3) “The paper lacks ablation studies for the different settings of hypernetwork component, leaving its individual contribution to the overall results unclear.”** In this paper we do perform ablation studies to separate the contribution of the hypernetwork from just conditioning on the state again. Additionally, we evaluate alternative initialization of the hypernetwork. However, we note that varying the size of the Hypernetwork was already ablated in (Beck et al., 2022), so we use the best size from that investigation. Thus, we feel we have properly covered the space of hypernetwork design. Most of this paper focuses on showing that a very simple recurrent hypernetwork (with few components to ablate) can outperform contemporary task-inference approaches. Thus, we spend the most effort ablating the design space of task-inference for comparison.
> * **4) “The format of the reference is wrong”** Could you elaborate on the reference format concern? As far as we are aware, according to the formatting instructions for 2023, “Citations may be author/year or numeric, as long as you maintain internal consistency. As to the format of the references themselves, any style is acceptable as long as it is used consistently.” Link: https://media.neurips.cc/Conferences/NeurIPS2023/Styles/neurips_2023.pdf
>
> **Questions:**
> * **1) “Why in the Cheetah environment, RNN+HN is worse than RNN+S?”** In Cheetah-Vel, RNN+HN has the same asymptotic return as RNN+S and higher initial returns at the start of training. For a brief period of time in the middle of training, RNN+S does achieve higher returns. Since this environment requires less action diversity than the rest (e.g. running at different velocities but not different directions), the curves can likely be explained as follows: In the beginning, RNN+HN learns behavior shared between the tasks more easily due to Bias-HyperInit that ignores the task, then RNN+S is able to learn the task adaptation marginally faster due to very small difference in actions between tasks, then RNN+HN catches up as it is just capable of learning small (and large) changes.
> * **2) “Please offer more ablation studies of the hypernetwork component.”** (Addressed above.)

---

> > ### Author Response · Authors · 2023-08-20
> >
> > Following up on the range of environments (1), we now have new experiments that show RNN+HN to match VI+HN on ML10, in addition to our MineCraft results that show superior performance. Please see the new global comments. As this seemed to be much of the prior concern in this review, if we have addressed this issue, please do consider adjusting the score accordingly.

---

### Official Review · Reviewer_NnYJ · 2023-07-09

**Soundness:** 3 good
**Presentation:** 2 fair
**Contribution:** 2 fair
**Rating:** 6
**Confidence:** 1

**Summary:**

The manuscript proposes to modify a common meta-RL baseline, in which a meta-learned RNN provides the policy network with an encoding of the trajectory. The proposed recurrent hypernetwork instead lets the RNN predict the weights and biases of the policy network. This is shown to be a strong model in comparison with several variations of RNN-based and task-inference meta-RL approaches.

**Strengths:**

I think the proposed approach is novel and the idea is described in sufficient detail.
The experimentation affords equal computational resources for hyperparameter optimization of all models. The method seems to perform better in 6 out of 7 considered environments.
The ablation studies adequately address some questions that I thought of while reading the paper.

**Weaknesses:**

1. The implementation of the RNN and RNN+HN models could've been described in a bit more detail. It is for example not clear what the meta parameters $\theta$ correspond to in the RNN baseline, since line 108 says that $f$ and $\pi$ use distinct parameters from $\theta$. Is this sentence wrong and $\theta$ are the RNN's parameters? I could also not find all hyperparameters of the architectures, such as the type and layer sizes of the RNNs.
1. For the strong claim that recurrent hypernetworks are SOTA in (all of) meta-RL, the presented experiments seem rather limited. What about experiments on RLBench or Meta-World? Also, there are many more categories of meta-RL methods to be considered than RNN-based and task-inference methods. The manuscript can benefit greatly from a clear justification of this strong claim.
1. I could not find an indication of how exactly the performance curves were generated, e.g. how many seeds per model, what exactly does the shaded area convey, etc.

**Questions:**

1. References to figures such as "See Figures 3 and 4" in line 123, I'd put in parentheses directly in the preceding sentence or say something like "See Figure X for a diagram of Y".

## Acknowledgement of rebuttal
I have read the rebuttal and reviews and engaged in discussion with the authors. The authors have mostly addressed my concerns by removing the general claims of the proposed method being state-of-the-art in meta-RL, by providing some clarification on the architecture and experimental setup, and by providing additional experimentation that support some of the design decisions. I have accordingly increased the score.

**Limitations:**

The authors mention that they cannot guarantee that the proposed method will improve over every baseline nor over every environment. This does not feel like a careful look at model details and how they might affect adaptation to different task variations. One general criticism of many meta-RL methods is that they are evaluated on very narrow task distributions. Maybe the authors could focus on that. The recent meta-RL survey by Beck et al. [1] might give some ideas for discussion.

## References
1. Beck, Jacob, et al. "A survey of meta-reinforcement learning." arXiv preprint arXiv:2301.08028 (2023).

---

> ### Author Rebuttal · Authors · 2023-08-08
>
> Thank you for your feedback. We especially appreciate your time understanding that this paper is not in your field. We appreciate you noting that the ablation studies answered many questions, that the proposed method is stronger on 6/7 of the environments, and that our hyperparameters optimization affords equal computation to all baselines. Responses to your feedback are below. If we have addressed these concerns, please do raise our score, and if not, please let us know what remains unclear:
> * **1) “The implementation of the RNN and RNN+HN models could've been described in a bit more detail. It is for example not clear what the meta parameters  correspond to in the RNN baseline, since line 108 says that  and  use distinct parameters from . Is this sentence wrong and  are the RNN's parameters? I could also not find all hyperparameters of the architectures, such as the type and layer sizes of the RNNs.”** We use a single layer RNN of size 256, as in Beck et al., 2022. We have added this to the appendix. Additionally, we have clarified in the main body that “$f$ and $\pi$ each use distinct subsets of the meta-parameters, $\theta$.” We meant that the parameters are distinct from each other, not distinct from $\theta$. Thank you for pointing out this ambiguity.
> * **2) “For the strong claim that recurrent hypernetworks are SOTA in (all of) meta-RL, the presented experiments seem rather limited. What about experiments on RLBench or Meta-World? Also, there are many more categories of meta-RL methods to be considered than RNN-based and task-inference methods. The manuscript can benefit greatly from a clear justification of this strong claim.”** For environments, we do agree that more environments would make our claim stronger and have done so. For details, please see the global response above. For baselines on the remaining experiments, we chose to ablate many task-inference methods to show that simpler black-box methods could outperform them. We believe we have sufficiently covered this space. As for PPG methods, it is already known that such methods cannot solve the benchmarks we use, since they are meant to test rapid adaptation within 1 and 2 episodes, which is not sufficient for PPG methods (Beck et al., 2023) and also shown to be worse in (Zintgraf et al., 2020). Moreover, we were simply looking to show that simpler end-to-end methods can beat the contemporary task-inference methods, not that they all will.
> * **3) “I could not find an indication of how exactly the performance curves were generated, e.g. how many seeds per model”** The learning curves show meta-episode return, optimized over five learning rates and averaged over three seeds, with a 68% confidence interval using bootstrapping. This information is in section 2 of the Appendix, but we have also now added it to the main body.
>
> **Questions:**
> * **1) “References to figures such as "See Figures 3 and 4" in line 123, I'd put in parentheses directly in the preceding sentence or say something like "See Figure X for a diagram of Y".”** We have updated these references to match your suggestion.

---

> > ### Comment · Reviewer_NnYJ · 2023-08-11
> >
> > Thank you for your response and providing some clarification.
> > I agree with reviewer YdBr that it is quite a big change to modify the claim, since the SOTA claim was prominently featured in many parts of the paper. Also, as brought up by reviewer YdBr and acknowledged by the authors in their response, my impression that the RNN+HN method is novel, was wrong (please correct me, if I misunderstood). I will increase the score slightly to a borderline reject, since the authors answered my questions and added a new supporting experiment. I will wait for further discussion with other reviewers to decide whether to adapt the score further. In the meantime, could the authors please indicate how hyperparameters for the new MineCraft experiment were chosen for each method?

---

> > > ### Author Response · Authors · 2023-08-11
> > > **Response to NnYJ**
> > >
> > > Thank you for your feedback. We appreciate you increasing the score and mentioning that you are open to adjusting it further. A response to the concerns you raise here are below. If this does not answer your questions, please let us know and we can clarify further:
> > >
> > >   * **1) "quite a big change to modify the claim"** Mostly what we just did was change *SOTA* to *surprisingly strong*. We would like to emphasize that this is NOT because we believe our method is not SOTA, but simply due to the difficulty of establishing SOTA, as noted by reviewers, when there is no agreed upon standard benchmark, other than MuJoCo, in meta-RL (Beck et al., 2023). Moreover, we feel strongly that even if our results were weaker and only showed RNN+HN to match baselines, it would still be of significant interest to the field. Given that it has widely been assumed in meta-RL that complicated task-inference methods are necessary to be competitive, this paper is the first to present strong evidence in the other direction. Given that we evaluate on MuJoCo, and now MineCraft, we feel strongly that it is of great interest to the meta-RL field that a black box can outperform much more complicated contemporary task-inference methods, some of which consider themselves SOTA (Beck et al., 2022).
> > >
> > >   * **2) "hyperparameters for the new MineCraft "** The learning rate was retuned, as for all environments over [3e-3, 1e-3, 3e-4, 1e-4, 3e-5]. Most other parameters were used from GridWorld, since it also has a discrete action space. Some parameters (e.g. the parameter in the VariBAD codebase called policy_num_steps) were taken from MuJoCo (generally Cheetah-Dir). While some others were chosen reasonably but without tuning (e.g. 2 inner-episodes to form a meta-episode and an action embedding of 8). We expect to release all parameters specific to the MineCraft environment when we release our code.

---

> > > > ### Author Response · Authors · 2023-08-20
> > > >
> > > > Following up on Meta-World (Weakness 2), we now have new experiments that show RNN+HN to match VI+HN on ML10 (in addition to our MineCraft results). Please see the new global comment. As this seemed to be much of the prior concern in this review, if we have addressed this issue, please do consider adjusting the score accordingly.

---

> > > > > ### Comment · Reviewer_NnYJ · 2023-08-21
> > > > >
> > > > > I thank the authors for the additional experimentation and providing some more details. I'll increase the score to a borderline accept

---

### Official Review · Reviewer_YdBr · 2023-07-13

**Soundness:** 2 fair
**Presentation:** 2 fair
**Contribution:** 2 fair
**Rating:** 4
**Confidence:** 3

**Summary:**

This paper is an empirical investigation that tests the performance of different variants of three methods for meta-RL: RNN that provides task information implicitly from previous trajectories on the same MDP, TI that trains a task representation with VAE manner, and VI that is the baseline Varibad [1]. On mujoco environment, the paper finds that RNN with output being a hypernetwork that outputs the parameter for the policy works the best, and thus re-establish [2] RNN+hypernetwork as the state-of-the-art method for meta-RL.

**References:**

[1] Luisa Zintgraf, Kyriacos Shiarlis, Maximilian Igl, Sebastian Schulze, Yarin Gal, Katja Hofmann, and Shimon Whiteson. Varibad: A very good method for bayes-adaptive deep rl via meta-learning. In International Conference on Learning Representation (ICLR), 2020.

[2] Jacob Beck, Matthew Jackson, Risto Vuorio, and Shimon Whiteson. Hypernetworks in meta-reinforcement learning. In CoRL, 2022.

**Strengths:**

1. Many variants are tested in the environment. There are 12 variants tested in the paper, which gives a thorough analysis on the effect of using hypernetwork for each method; in addition, the effect of "conditioning on the state twice" is also considered, which makes the conclusion that hypernetwork is crucial more rigorous.

2. The figures appended in the paper does a well job in conveying the architecture of each method.

**Weaknesses:**

**1. The experiment results are not convincing enough.**

a) The method proposed (line 184) in the paper, RNN+hypernetwork, is already proposed in prior work [1]. More specifically, RNN alone is equivalent to RL2 [3] as line 106 suggests, and RL2+hypernetwork is already tested in [1], which is inferior to Varibad [2]+hypernetwork. Thus, the key contribution of this paper is not proposing new method, but to overthrow the previous conclusion that Varibad+hypernetwork is better. However, this paper only tests gridworld and mujoco environments, and on both environments RNN+HN is only marginally better than VI+HN (which in [1] also has multiple methods that works similarly well, indicating that mujoco alone is not a strong benchmark). Most importantly, the metaworld (ML1 and ML10) environment, which is the decisive evidence that Varibad+hypernetwork is better than RL2+hypernetwork, is missing in this paper. With such result missing, it is hard for the readers to be convinced that the result is the other way around from that in [1].

b) The baselines, though with many variants, do not cover enough areas for the claim of the state-of-the-art. Reviewers of [1] have already pointed out that the state-of-the-art status of Varibad [2], even before the presence of [1], is questionable. Also, while it is true that the meta-RL method can be briefly summarized in policy gradient, implicit task representation and task inference as the paper suggests (line 61-63), in Table 2 of [4] there are many branches within each of the direction. For example, what if transformer instead of RNN is used [5]? Is policy gradient method necessarily worse than the other two branches? For a state-of-the-art method, those methods also need to be considered.

**2. The delivery of the paper's idea can be improved.**

a)  There is no motivation stated about why hypernetwork is used. The paper only states that "we present the key insight that the use of a hypernetwork architecture is crucial ..." (line 36-37) and "hypernetwork has never been widely evaluated in meta-RL" (line 81-82). However, there is no intuitive explanation about why hypernetwork is useful for meta-RL and should be considered in the first place. While there is explanation in [1] about preventing degeneration of multi-tasking, the paper should be self-contained and inform the readers about why hypernetwork, the most important component in the finally proposed RNN+hypernetwork method, is considered for meta-RL.

b) The method section can be modified to convey the idea more clearly. Currently, it is hard to identify which method is proposed (line 184) by the paper, and it is unclear for the first-time readers about why multiple methods are listed in the method section (and most of them is not proposed by this paper), instead of the usual paradigm where each key component is listed in a subsection. Since the methods are not novelly proposed by this paper, they can be put into experiment setup section (with more emphasis that this is an empirical study paper rather than conventional paper that proposes a new method, e.g., add a contribution summary at the end of introduction section); or, a summary could be added in the front of the method section, to tell the readers that all those methods listed below are tested in the experiment section, and in each section we discuss one type of method tested. A table summary would also be very helpful.

c) There is no detail of the environment settings except gridworld. What is the cheetah-dir environment? What are the definitions of state, action and reward, and how are the MDPs in meta-learning differs? What reward can be considered expert-level performance? While they can be found in prior work, the paper should be self-contained and the settings should be attached in the appendix to help the reader form a better intuition about the environment.

**3. Other minor problems:**

a) Some experiment results are missing, for example, RNN in Fig. 8b, 8c, 9b, 9c and 9d;

b) Fig. 10 is never referred to in the paper; reference to Fig. 10 should be added in "latent gradients" paragraph.

c) The x-axis of walker environment is not aligned with other mujoco environments.

d) The name, VariBad, should be mentioned in VI and VI+HN to more clearly show the connection of the method tested to existing baselines.

**References:**

[1] J. Beck et al. Hypernetworks in meta-reinforcement learning. In CoRL, 2022.

[2] L. Zintgraf et al. Varibad: A very good method for bayes-adaptive deep rl via meta-learning. In ICLR, 2020.

[3] Y. Duan et al. $RL^2$: Fast reinforcement learning via slow reinforcement learning. In arXiv, 2016.

[4] J. Beck et al. A survey of meta-reinforcement learning. In arXiv, 2023.

[5] L. Melo. Transformers are Meta-Reinforcement Learners. In ICML, 2022.

**Questions:**

Besides those listed in the weakness section, I have one question: why the performance of RNN+HN in Fig. 9a and Fig. 10a is different?

Below are my suggestions for this paper:

1. Modify the introduction and method section of the paper such that it is more clear that this is an empirical study instead of proposing a novel method, and add contribution summary at the end of the introduction. Add a table that summarizes the methods tested in the paper.

2. Add environment details of the mujoco environments in the appendix.

3. Compare RNN+HN to more methods summarized in [1];

4. Compare RNN+HN with VI+HN (and other baselines) on metaworld and other more challenging environment;

5. Fix other minor issues mentioned in the weakness section.

6. I strongly advice the authors to open source their code upon acceptance / next submission. The author could also consider adding license statement in the appendix.

**References:**

[1] J. Beck et al. A survey of meta-reinforcement learning. In arXiv, 2023.

**Limitations:**

There is an independent limitation section in the paper, which I think generally discusses an important limitation of the work. Though, as suggested in the weakness section, I think currently the limitation is not sufficiently mitigated. There is no negative societal impact discussed in the work, which I would encourage the readers to add; though the work is still far from application, automated control itself brings potential challenge, such as job loss, to the human society.

---

> ### Author Rebuttal · Authors · 2023-08-08
>
> Thank you for your feedback. We appreciate you noting the many benchmarks that we compare against and the rigorous conclusion that the hypernetwork is critical. We address your concerns below. Specifically, we add experiments (by running RNN on more domains and adding an additional domain) and add details (motivation for the hypernetwork and method details), as requested. Could you please raise the score correspondingly, or let us know if any topic remains unclear?
> * **1) Experimental Results**
>   * **a) Meta-World and Degree of Improvement** We do agree that more environments would make our claim stronger and have run more experiments. For this evidence, please see the global response above. Additionally, we also do feel strongly that our existing results show that RNN+HN is not just marginally better than HN+VI, but much better (and much simpler). For example, see the learning curves (blue vs orange) on Walker, Cheetah, and all of the gridworlds. Finally, there is no decisive evidence in the other direction from [1]: The results are not statistically significant in their table, and if you look at the appendix, the learning curves are highly unstable and difficult to draw conclusions from. This lack of evidence and baselines is the main reason for our paper.
>   * **b) Baselines** We have now adjusted all of our claims throughout the paper to assert that recurrent hypernetworks are surprisingly strong instead of SOTA. For our baselines, we chose to ablate a large number of task-inference methods to show that simpler black-box methods could outperform them. We believe we have sufficiently covered this space. As for PPG methods, it is already known that such methods cannot solve the benchmarks we use, since they are meant to test rapid adaptation within 1 and 2 episodes, which is not sufficient for PPG methods [1] and also shown to be worse in [2]. We do intend transformers with hypernetworks as future work, but which sequence model the hypernetwork conditions on is an orthogonal problem. Moreover, we were simply looking to show that simpler end-to-end methods can surprisingly beat the contemporary task-inference methods, not that they all will.
> * **2) Delivery**
>   * **a) Hypernetwork Intuition** We agree that we can make the paper more self-contained by adding motivation for hypernetworks as in [1] and have now done so. In addition to preventing interference between different tasks and scaling trends suggested by Beck et al., 2022, we hypothesize that the low norm afforded by our models, i.e., low  sensitivity to the latent variable, and re-conditioning on state are crucial for stable training. We have added some text to expand on existing motivation for the use of hypernetworks in related work and have added some text on our analysis in the introduction to make this paper more self-contained.
>   * **b) Empirical Study, Contribution Summary in Introduction, Methods Table** We agree that this could be confusing and have added clarification to the introduction and methods to emphasize that this is an empirical study and RNN+HN is not a novel method, though it is the one we find gives the strongest results, and we add a table comparing method components to the appendix. Please see the global response above.
>   * **c) MuJoCo Details in Appendix** We have added details on the MuJoCo environment to the appendix as requested. See the global response above for details.
> * **3)  Minor problems**
>   * **a) Missing Results** We ran these experiments and results show that the RNN in fact always performs the same or worse than RNN+S and RNN+HN, as expected. Plots can be seen in the global response above.
>   * **b) Fig. 10 Reference** The reference has been added.
>   * **c) X-axis of Walker** We have run the limiting Walker experiment for longer and updated the plot with no significant changes to the results. Plots can be seen in the global response.
>   * **d) Calling VariBAD by Name** We have also now added the name Varibad, as requested.
>
> **Questions:**
> * **0) Performance Difference Between Figures** The difference in curves should be due to different seeds for each plot.
> * **1) Empirical Study, Contribution Summary in Introduction, Methods Table** We have added clarification to the introduction and methods to emphasize that this is an empirical study and not a novel method. We have likewise updated the contribution paragraph at the end of the introduction and created a table (see the global response above).
> * **2) MuJoCo Details in Appendix** We have added details on the MuJoCo environment to the appendix. The details can also be seen in the global response above.
> * **3) Methods from [1]** We do compare to the baselines in [1]. In [1], the authors only compare to VariBAD [2] and RL2 [3]. We added the many baseline variants that you mentioned precisely in order to address this limitation in [1]. Note that [1] Does evaluate VariBAD with many different initializations for the hypernetwork and FiLM, but [1] shows them all to be inferior, which we have also found to be true in our experience. (For other types of methods, see 1b above.)
> * **4) Challenging Environments** The comparison on Meta-World was already made by [1] and the results are not statistically significant in the table. If you look at the appendix in [1], the learning curves are highly unstable and difficult to draw conclusions from. For this reason, we chose to supplement our experiments with MineCraft from AMRL (Beck et al., 2020) instead. See the the global response for details.
> * **5) Minor Issues** (Addressed above.)
> * **6) Code** We plan to release code.

---

> > ### Comment · Reviewer_YdBr · 2023-08-11
> > **Response to the Rebuttal**
> >
> > Thanks for your detailed response, and I appreciate the efforts made by the authors to address my concerns. I think concerns such as delivery are well-addressed. Here are my follow-up responses:
> >
> > 1. While I agree that the supplementary minecraft environment is a stronger evidence than that shown in the original paper that RNN+HN is better than HN+VI, for the claim of "much better" in RL, I would expect at least one scenario where RNN+HN succeeds but HN+VI only reaches medium-level performance (e.g. 50-70% reward) or even completely fails. The performance difference on gridworld and mujoco only seem to be either an issue of convergence speed of ~10% performance difference.
> >
> > 2. Regarding the result for ML1/ML10 in [1]:
> >
> > a) I am referring to the Table 2 of [1] for result (on the 7th page on ArXiv version), where on ML10 task, the performance comparison between VariBAD+HN and RL2+HN is as follows:
> >
> > -|VariBAD|RL2
> > ---|---|---
> > HFI|**$28.4\pm 6.0$**| $7.1\pm 2.4$
> > Bias-HyperInit|**$23.9\pm 6.2$**|$14.2\pm 7.2$
> >
> > b) in line 106, the authors write "**RNN** ... is equivalent to RL2"; in line 112, the authors write "we follow the initialization method for hypernetworks, Bias-HyperInit, suggested by Beck et al.". Thus, I assume that RNN+HN, the best method found in this paper, is equivalent to the RL2+Bias -HyperInit, which has the performance of $14.2\pm 7.2$ as opposed to $23.9\pm 6.2$ with VariBAD. When you plot them on a figure (which is the bottom-right figure of Figure 1 in the appendix, on the 13th page of the ArXiv version), even with large variance, the upper edge of the RNN+HN can barely touch the lower edge of VariBAD+HN, and the reward is almost doubled in VariBAD. Thus, I think this can be called significant despite the large variant.
> >
> > 3. Another thing that worries me from giving a higher score is that the change of claim is a major revision to the paper's contribution and organization.
> >
> > To sum up, I tend to keep my scores for now and see how the discussion goes (both ours and the discussion between the author and the other reviewers).
> >
> > **References:**
> >
> > [1] J. Beck et al. Hypernetworks in meta-reinforcement learning. In CoRL, 2022.

---

> > > ### Author Response · Authors · 2023-08-11
> > > **Response to YdBr**
> > >
> > > Thank you for the detailed response. We believe there was still some misunderstanding that we have resolved below. Please let us know if this addresses your concerns, and if not, then what else we can clarify:
> > >
> > >   * 1 **"I would expect at least one scenario where RNN+HN succeeds but HN+VI only reaches medium-level performance (e.g. 50-70% reward) or even completely fails."** In terms of area under the curve, our results do show significantly better performance. In terms of the baselines failing, our new MineCraft results does actually show this. Note that the scale of the rewards can be misleading. For example, while the performance difference on Minecraft is only about 25%, RNN+HN can solve the task while HN+VI cannot. The reason the reward difference is not larger is that the (easy) dense reward for maze navigation, which is unrelated the test of adaptation using memory, happens to represent the bulk of the reward obtained. HN+VI does truly fail the task here. Moreover, we feel strongly that even if our results were weaker and only showed RNN+HN to match baselines, it would still be of significant interest to the field. Given that it has widely been assumed in meta-RL that complicated task-inference methods are necessary to be competitive, this paper is the first to present strong evidence in the other direction.
> > >
> > >   * 2 **" I am referring to the Table 2 of [1] ...I think this can be called significant despite the large variant."** In [1], the authors actually did run "two-tailed t-tests with p = 0.05 to determine significance" and put in bold results significantly different from the rest. No results on ML10 were significantly different. This contributed to our hesitancy to rely on ML10.
> > >
> > >   * 3 **"major revision to the paper's contribution and organization."** Mostly what we just did was change *SOTA* to *surprisingly strong*. We would like to emphasize that this is NOT because we believe our method is not SOTA, but simply due to the difficulty of establishing SOTA when there is no agreed upon standard benchmark, other than MuJoCo, in meta-RL (Beck et al., 2023). Given that we evaluate on MuJoCo, and now MineCraft, we feel strongly that it is of great interest to the meta-RL field that a black box can outperform (or even match) much more complicated contemporary task-inference methods, some of which consider themselves SOTA (Beck et al., 2022).

---

> > > > ### Author Response · Authors · 2023-08-20
> > > >
> > > > Following up on Meta-World (2), we now have new experiments that show RNN+HN to match VI+HN on ML10. Please see the new global comment. As this seemed to be much of the prior concern in this review, if we have addressed this issue, please do consider adjusting the score accordingly.

---

> > > > > ### Comment · Reviewer_YdBr · 2023-08-21
> > > > > **Further Response**
> > > > >
> > > > > Thanks for the reviewer's response; after reading the other reviewers' reply, I decide to change my score from 3 to 4 to reflect my appreciation to the authors' efforts on addressing my concerns. I am still leaning on rejection as I am not fully convinced by the reviewer on the significance of performance gain and concern of claim change; for example, not many papers run significant tests in the RL community to decide whether the result is significantly better or not, and from my judgment, I still feel that **$23.9\pm 6.2$** over **$14.2\pm 7.2$** is (usually viewed as, though not rigorously defined as) significant difference.

---

> > > > > > ### Author Response · Authors · 2023-08-21
> > > > > >
> > > > > > Thank you for the score adjustment. It is much appreciated. It seems like this is the main sticking point, so if it helps to convince you further, we can provide additional information.
> > > > > >
> > > > > > We are in contact with the original authors. The original results (over three seeds) for RNN+HN were: 11.31, 3.37, 27.77. The results for VI+HN, including our new results with their code, are: 12.48, 25.69, 33.61, (32.80, 3.32, 18.93), where the latter three results are our seeds on their code. At the very least, their result is not strong. And this singular result, if anything, suggests the need for the much stronger evaluation in our paper.
> > > > > >
> > > > > > Additionally, with our experiments, the curves and confidence intervals on ML10 actually overlap entirely. (The interval for RNN+HN lies entirely within that of VI+HN.) Moreover, the mean for RNN+HN is in fact higher than VI+HN. Thus, at the very least, their result is not reproducible.

---

### Author Rebuttal · Authors · 2023-08-08

In order to address reviewers requests, we have added additional experiments (environments and baselines), adjusted claims, and added details (in a table and text). Figures for new experiments are in an attached PDF, and specifics are below.

**TLDR:**
* We have removed claims to SOTA
* New experiments in MineCraft corroborate our claims
* New experiments on previous domains corroborate that RNN alone is a weak baseline
* We have added environment details and method details and motivation in text and a table

**Updated Title and Claim:**
Some reviewers felt that more experiments were needed to claim SOTA. To address this feedback, we have added experiments and removed claims to SOTA. For instance, our new title is *Recurrent Hypernetworks are Surprisingly Strong in Meta-RL*, and we have clarified our contributions at the end of the introduction. Here, we emphasize that we conduct an empirical investigation showing the value of hypernetworks in maximizing the potential of recurrent networks. We clarify upfront that while the use of a hypernetwork with RNNs is not a novel idea, they have never been evaluated in meta-RL beyond a single environment, let alone shown to outperform contemporary task-inference methods (Beck et al., 2022).

**Additional MineCraft Environment:**
In addition to toning down our claims, we have added more experiments to demonstrate the strength of RNN+HN on more complex and challenging domains. Some reviewers have suggested Meta-World and Alchemy. Meta-World is more challenging; however, such benchmarks are generally too difficult to solve (Beck et al., 2023) and high variance, making them extremely difficult to get informative comparisons even after weeks of training (Beck et al., 2022). Similar issues exist for Alchemy (Beck et al., 2023). Instead, we add experiments with a MineCraft environment.

Specifically, we supplement our results with experiments in the MC-LS environment from AMRL (Beck et al., 2020), since it can easily be adapted to meta-RL and was originally designed to test more challenging long-term memory from visual observations in MineCraft. In this environment, the agent must navigate through a series of 16 rooms. In each room, the agent must navigate left or right around a column, depending on whether the column is made of diamond or iron. Correct behavior receives a reward of 0.1. Finally, at the end, the agent must move right or left depending on a signal (red or green) that was shown before the first room. Correct behavior receives a reward of 4, while incorrect behavior receives a reward of -3. In our experiments, we allow the agent to adapt over two consecutive episodes, forming a single meta-episode, where each direction corresponds to a task.

On MC-LS we compare RNN+HN to VI+HN, since VI+HN is an established contemporary task-inference baseline (Beck et al., 2022). Additionally, we add an additional seed (four in total) and a linear learning rate decay due to high variance in the environment. In the MineCraft Figure in the PDF, we see that RNN+HN significantly outperforms VI+HN. While VI+HN learns to navigate through all rooms, it does not learn the correct long-term memory behavior. In contrast, RNN+HN is able to adapt reliably within two episodes, and one seed even learns to adapt reliably within a single episode. While further work is needed to learn the optimal policy, these experiments demonstrate that RNN+HN outperforms VI+HN, even on more challenging domains.

**Additional RNN Baselines:**
Some reviewers noted that the RNN baseline was not run on several MuJoCo experiments and that some results on Walker were not run as long. This was due to computational limitations. However, we have now added the RNN baseline to all MuJoCo experiments, and we have run all methods on the Walker experiment longer to match the duration of the other MuJoCo environments. Please see the PDF for figures. Results are consistent with previous claims: RNN is a weak baseline, RNN+S is a stronger baseline, RNN+HN outperforms both.

**Methods Table:**
A table summarizing the components in each all baseline methods has been added to the appendix:
| |Inference Target|Policy Conditions on State|Hypernetwork|Inference Pre-Training and Parameter Reuse|
|-|---|---|---|---|
|RNN|None|No|No|N/A|
|RNN+S|None|Yes|No|N/A|
|RNN+HN|None|Yes|Yes|N/A|
|TI Naive|Given|Yes|No|N/A|
|TI|Learned|Yes|No|No|
|TI++|Learned|Yes|No|Yes|
|TI+HN|Learned|Yes|Yes|No|
|TI++HN|Learned|Yes|Yes|Yes|
|VI|Transitions|Yes|No|N/A|
|VI+HN|Transitions|Yes|Yes|N/A|
|BI++HN|Base Net|Yes|Yes|N/A|

**MuJoCo Details:**
One reviewer noted that details on the MuJoCo environments should be included in order to make the paper more self-contained. We have added these details to the appendix. We share the details here for two environments (Cheetah-Dir and Walker) as examples.

We evaluate on all four MuJoCo environments used by Zintgraf et al., 2020. All environments require legged locomotion.

In Cheetah-Dir, the agent controls a robotic cheetah morphology by outputting a control torque for each of six joints on the morphology. The task is to run forward or backward with as large a velocity as possible. The agent observes the position, angle, and velocity of each body part (17 dimensions in total) and is given a positive reward for its velocity in the direction given by the task and a negative reward for control costs (specifically, 5% of the magnitude of the action vector).

In Walker, the agent controls a two-legged torso morphology and outputs a control torque for each of six joints on the morphology. The observations are 17 dimensional for this environment. The tasks are defined as uniform samples of 65 different physics coefficients (e.g. body mass and friction) for the simulation. The agent is given a positive reward for the forward velocity, a positive reward of one reward per timestep, and a negative reward for the control costs (specifically, 0.1% of the magnitude of the action vector).

---

> ### Author Response · Authors · 2023-08-20
> **New Meta-World Experiments**
>
> **TLDR:** New Meta-World experiments show RNN+HN matches performance of VI+HN on ML10, which should address the last major concern of the reviews.
>
> Some reviewers have requested ML10 (from Meta-World), noting that Beck et al., 2022 demonstrated RNN+HN to underperform VI+HN on ML10. (This is the only experiment ever run with RNN+HN in meta-RL prior to our paper.) We noted that Beck et al., 2022 actually did run "two-tailed t-tests with p = 0.05 to determine significance" and found NO results on ML10 to be significantly different, which makes the result a bit misleading. The fact that that singular results carries so much weight is one of the reasons that the much stronger countervailing evidence in our paper is so necessary to the conversation.
>
> However, to allay the concerns of reviewers, we have conducted this experiment. While we are unable to upload an additional PDF, we will describe the results here and include the plots with the final paper. First, we find that the learning curves for RNN+HN and VI+HN overlap consistently throughout training. For instance, on training curves that run from 0 to 7e7 frames, the curves intersect near training steps 2e7, 4e7, 5e7, and 7e7. Second, we find that the variance of RNN+HN is significantly less than that of VI+HN. For nearly all of training, the confidence interval of RNN+HN falls within that of VI+HN. And finally, the final return of RNN+HN is higher (thought not significantly higher) than VI+HN. RNN+HN has a final average test success percent of 17.22, compared to 13.87 for VI+HN.
>
> While these results are not sufficient to show superiority on ML10, we do show superiority on other environments in the paper. Additionally, even matching existing task-inference baselines would still be of significant interest to the field. Given that it has widely been assumed in meta-RL that complicated task-inference methods are necessary to be competitive, this paper is the first to present strong evidence in the other direction.
>
> Finally, to ensure that the difference from Beck et al., 2022 is legitimate, we acquired code from that paper and re-ran that experiment (which runs marginally longer and with a smaller latent encoding size). We only had sufficient compute to re-run VI+HN. Still, using different seeds from the original paper resulted in worse performance, indicating that the results truly were insignificant due to variance. Specifically, the final mean return is 18.35 for VI+HN, with is an average over three seeds, with final returns 32.80, 3.32, 18.93, respectively. In contrast, the original paper reports a mean return of 23.9 for VI+HN. Clearly, with such high variance the comparison between these two methods on ML10 is not useful, hence our reluctance to use ML10 and our additional MineCraft experiments.
>
> We hope that this additional evidence on the requested benchmark puts all questions of ML10 to rest.

---

### Decision · Program_Chairs · 2023-09-21

**Decision:**

Accept (poster)

**Comment:**

Initially, the paper received mixed scores. The reviewers liked the idea of revisiting hypernetwork for meta-RL tasks and noted the investigation of different variants of the proposed method. However, they were concerned about the limited evaluation environments, unconvincing performance improvement, and a lack of comparison with some baselines, which cannot support the strong claim of the proposed method being state of the art in meta-RL. Additionally, they were concerned about the limited novelty (being an empirical study of an existing method), the unclear motivation, the absence of explanation regarding some important design choices, implementation details, and ablation studies, and the quality of the writing and presentation.


The rebuttal dealt partially with the reviewers’ points by toning down the state-of-the-art claim to be surprisingly strong, and offering additional explanation, experimental results (including more baselines and benchmarks, such as some Meta-World experiments), and discussion.


After rebuttal, Reviewers rCPG, NnYJ, and 8GEx rated the weak accept or accept score, Reviewer k7Er rated the borderline accept score, and Reviewer YdBr rated the borderline reject score. In the subsequent discussion, Reviewer YdBr continued to express concerns regarding the major change of the paper’s claim and marginal performance improvement in experimental results, but Reviewer YdBr did not insist on rejecting the paper. On the other hand, Reviewers NnYJ and 8GEx championed the paper, pointing out that the changes made during the rebuttal can be incorporated into the camera ready version without the need for another review cycle.


The AC has read the paper, reviews, and rebuttal, and discussed with the reviewers at length. The AC found the arguments of Reviewers NnYJ and 8GEx to be the most persuasive. While the experimental evaluation did not support the paper’s original claim of state-of-the-art performance in meta-RL, the conducted empirical study still demonstrated interesting findings and competitive performance, which could provide a different perspective and useful basis to develop simpler meta-RL methods for future work in this direction. Also, during rebuttal, the authors have committed to removing this claim and revising the title and narrative accordingly in the revision. These changes would not lead to a significant rewriting of the paper.


The authors are encouraged to improve the camera ready version by following reviewer recommendations, in particular precisely revising the claim and contributions as promised (i.e., not state of the art, being an empirical study rather than proposing a novel method), including new experimental results provided in the rebuttal and more in-depth discussion of limitations. While some results were incomplete due to time and computational constraints during rebuttal, it would be beneficial to conduct the full set of experiments and incorporate them in the camera ready version. This work would greatly benefit from such revision and its impact can be significantly improved. This decision was discussed with and approved by the AC and SAC.